# TOWARDS INTERPRETABLE CONTINUAL LEARNING THROUGH CONTROLLING CONCEPTS

## ABSTRACT

Continual learning is a challenging task in machine learning as models can learn new tasks easily but suffer from catastrophic forgetting of previous tasks. In this work, we propose a novel framework called **Concept Controller** that addresses the issue of catastrophic forgetting by systematically controlling interpretable concepts in deep neural networks. Our method has several advantages: (1) *High Performance*: empirical results show that our method outperforms exemplar-free methods and is comparable with exemplar-based methods in the standard metrics such as average accuracy and average forgetting. Moreover, combining our method with exemplar-based methods can further improve the performance of exemplar-based methods. (2) *Light*: our method does not need extra memory space to store previous tasks' samples unlike the exemplar-based methods. (3) *Interpretable*: the procedure of controlling concept units is transparent.

## 1 INTRODUCTION

Continual learning is an essential aspect of machine learning, that allows models to adapt their behavior over time as new data becomes available. However, one major challenge of continual learning is "catastrophic forgetting" phenomenon, where previously learned knowledge is lost in the model after learning new tasks. This is mainly due to the distribution shift of inputs as the tasks change. Continual learning has three common settings (van de Ven et al., 2022; De Lange et al., 2021): class incremental setting, task incremental setting, and domain incremental setting. In this paper, we focus on the class incremental setting, which is the most challenging setting among all in continual learning that exhibits serious catastrophic forgetting behavior (van de Ven et al., 2022; Chaudhry et al., 2018a; De Lange et al., 2021).

Existing approaches for continual learning mostly belong to three categories: **(i)** regularization-based methods, **(ii)** architecture-based methods, and **(iii)** replay-based methods. The key idea of the regularization-based methods (Kirkpatrick et al., 2017; Zenke et al., 2017; Li & Hoiem, 2017) is to constrain the modification of important model parameters from previous tasks, while architecture-based methods (Rusu et al., 2016; Yoon et al., 2017) modify the model's architecture or parameters when learning new tasks. For replay-based methods (Lopez-Paz & Ranzato, 2017; Rebuffi et al., 2017), they utilize replay buffers to store previous tasks' information to update models. Category **(i)** and **(ii)** are also known as **exemplar-free methods**, which do not replay data from old tasks; while category **(iii)** is known as **exemplar-based methods**, which store previous data to replay. In general, replay-based methods have better performance but require extra memory storage. While these methods help to alleviate the issue of catastrophic forgetting, their performance is still far from satisfactory. Moreover, these methods are not interpretable, which hinders the development of better algorithms. Ideally, a more systematic and interpretable way to design methods is desirable.

On the other hand, there is an orthogonal line of work aim to understand the role of neurons in neural networks (Bau et al., 2017; 2020; Oikarinen & Weng, 2022; Hernandez et al., 2022; Mu & Andreas, 2020). If a neuron is highly related to a human-understandable concept, then it can be denoted as a "concept unit". In the transfer learning setting, previous work (Bau et al., 2017) discovered a positive correlation between the number of concept units and the classification accuracy at the target task, which indicates that the existence of task-related concept units improves model's classification accuracy. Since continual learning and transfer learning share many similarities, we expect that carefully manage and control task-related concept units will also improve continual

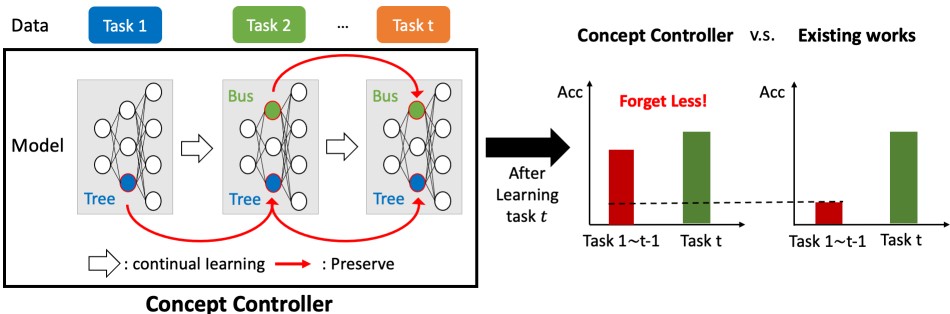

Figure 1: Overview of the proposed **Concept Controller**.

learning performance. Meanwhile, we find out that existing continual learning algorithms don't really preserve human-understand concepts in continual learning, as Table 3 shows. Building upon these insights, we propose a novel and interpretable framework called **Concept Controller** (CC) to mitigate catastrophic forgetting in continual learning by systematically manage and control the task-related concept units.

In this paper, our goal is to bring interpretability to continual learning, which allow us to improve the "method design" in an interpretable manner. Our method enables models to retain knowledge from previous tasks by effectively controlling the concepts learned from the tasks. Different from existing works in continual learning which are not interpretable, we aim to systematically control the human-understandable concepts retained in the model to mitigate catastrophic forgetting. With **Concept Controller**, we show that it is possible to preserve the learned knowledge from the previous tasks effectively – we outperform existing continual learning methods by up to 1.4% in average incremental accuracy. In addition, we propose a new Concept Bottleneck Model (Koh et al., 2020) for continual learning called **Concept Controller CBM** (CC-CBM). Our approach can convert any neural network backbone to retain concept units related to previous tasks while adding new concept-neurons related to the new tasks. We show that our technique can further reduce the average incremental forgetting by up to 9.1% compared to existing approaches. In summary, CC belongs to category **(i)**, and CC-CBM belongs to category **(ii)**. We demonstrate that our methods can improve category **(iii)** when combined with them. This shows that our methods are broad and comprehensive. Finally, our proposed methods have the following benefits:

- *High Performance*: Extensive experiments show that **Concept Controller** outperforms exemplar-free methods by up to 1.4% in average incremental accuracy and is comparable with exemplar-based methods in terms of average incremental accuracy and the average incremental forgetting metric (Zhou et al., 2023; Zhu et al., 2022). Moreover, combining **Concept Controller** with exemplar-based methods can further improve the performance of exemplar-based methods by up to 6.7% in average incremental accuracy.

- *Light*: **Concept Controller** does not need extra memory space to store previous tasks' samples unlike the exemplar-based methods.

- *Interpretable*: **Concept Controller** is transparent and allows us to understand and further control the knowledge in the model retained from previous tasks.

## 2 BACKGROUND AND RELATED WORK

Continual learning aims to learn the model's parameters $\theta$ for a series of tasks. Let $\mathcal{D}_t = \{(x_i^t, y_i^t)\}_{i=1}^{N_t} \in (\mathcal{X}^t, \mathcal{Y}^t)$ be the data for task $t$, $N_t$ is the number of data samples, $x_i^t \in \mathbb{R}^d$ is the data sample, and $y_i^t$ is the class label. In the class incremental setting, tasks don't share the class labels, which means $\mathcal{Y}^t \cap \mathcal{Y}^{t'} = \emptyset, \forall t \neq t'$. In the task incremental setting, tasks may share class labels, and every test sample's task label $t$ is provided in the testing phase. In the domain incremental setting, tasks have the same possible outputs, which means $\mathcal{Y}^t = \mathcal{Y}^1, \forall t$. In this work, we focus on the class-incremental setting, which is the most challenging setting (van de Ven et al., 2022; Chaudhry et al., 2018a; De Lange et al., 2021).

## 2.1 CONTINUAL LEARNING

To mitigate catastrophic forgetting in continual learning, several methods have been proposed. First, **(i)** regularization-based methods (Kirkpatrick et al., 2017; Zenke et al., 2017; Li & Hoiem, 2017; Jung et al., 2016; Dhar et al., 2019; Castro et al., 2018; Hu et al., 2019; Lee et al., 2019; Aljundi et al., 2018; Chaudhry et al., 2018a; Lee et al., 2017; Schwarz et al., 2018) add additional terms in the loss function to constrain model parameters to not change too much from previous tasks. Second, **(ii)** architecture-based methods (Rusu et al., 2016; Yoon et al., 2017; Xu & Zhu, 2018; Yan et al., 2021; Li et al., 2019; Serra et al., 2018; Wang et al., 2021; Zhu et al., 2022) modify the model's architecture or parameters when learning new tasks, by dynamic expansion or pruning. Third, **(iii)** replay-based methods (Lopez-Paz & Ranzato, 2017; Rebuffi et al., 2017; Chaudhry et al., 2018b; Rolnick et al., 2019; Hou et al., 2019; Wu et al., 2019; Buzzega et al., 2020; Wang et al., 2022; Guo et al., 2022; Liu et al., 2021; Aljundi et al., 2019) store previous tasks' information and train the model with new tasks jointly. The detailed introduction of **(i)** and **(iii)** is in Appendix A.3. Meanwhile, some works focus on the theoretical aspect of continual learning (Peng et al., 2023; Peng & Risteski, 2022; Cao et al., 2022; Ruvolo & Eaton, 2013; Pentina & Urner, 2016; Chen et al., 2022; Kim et al., 2022). However, none of these methods is directly controlling human-interpretable concepts, which makes them lack interpretability. Recent work (Marconato et al., 2023) design a framework to control neuro-symbolic concepts for neuro-symbolic continual learning. However, their method is only suitable to certain model architectures like DeepProbLog (Manhaeve et al., 2018) and datasets with predefined concepts like CLE4EVR described in their paper. In contrast, our methods are suitable to any CNN-based models and standard benchmark datasets. Another recent works (Rymarczyk et al., 2023) connects interpretability with continual learning. The method focuses on part-based prototype concepts. Our work focuses on text-based concepts instead, which allows more general interpretability. Meanwhile, it is only suitable for particular model architectures whereas our methods are suitable for any CNN-based models.

The representative techniques in architecture-based methods category **(ii)** is Dynamic Expandable Network (DEN) (Yoon et al., 2017), which is described in Appendix A.3. Compared with DEN, our method does not require retraining when learning new tasks. Moreover, our method is interpretable as we identify neurons with human-understandable concepts, allowing a better understanding of the retained knowledge from previous tasks. Indeed, interpretability is one of the main differences between our methods and existing architecture-based methods in continual learning.

## 2.2 NEURON-LEVEL INTERPRETION AND CONCEPT BOTTLENECK MODELS

Several works (Bau et al., 2017; 2020; Oikarinen & Weng, 2022; Hernandez et al., 2022; Mu & Andreas, 2020) provide automated descriptions of the roles of individual neurons in deep vision models, and do extensive studies for their methods' interpretability. Typically these methods generate a description by analyzing what kinds of inputs result in high activations for the given neuron. For example, Network Dissection (Bau et al., 2020) identifies the concepts of individual neurons by comparing the neuron's activation map to concept annotated data. A more recent work CLIP-Dissect (Oikarinen & Weng, 2022) eliminates the need of concept annotated data by leveraging the Contrastive Language-Image Pre-training (CLIP) model (Radford et al., 2021) and several similarity functions. Our proposed method works with both of them, but for efficiency we use CLIP-Dissect through out the experiments.

Concept Bottleleck Model (CBM) (Koh et al., 2020) has a layer called Concept Bottleleck Layer (CBL) where each neuron corresponds to a human interpretable concept. Recent works (Oikarinen et al., 2023; Yuksekgonul et al., 2022) try to address the problem that CBMs require training datasets with concept annotations which maybe expensive and hard to collect. Specifically, (Oikarinen et al., 2023) proposed Label-Free Concept Bottleneck Model (LF-CBM) to transform neural networks into an interpretable CBM without labeled concept data. The procedure in LF-CBM is as follows: First, it uses GPT-3 (Brown et al., 2020) to generate a set of text concepts important for the task based on class labels, which is then filtered to improve quality. Second, LF-CBM learns a CBL where each neuron corresponds to one of these concepts, by aligning the neurons with CLIP's(Radford et al., 2021) representation of the concepts. Given M concepts generated from the previous step, the CBL is a linear transformation of the pretrained NN backbone $f(x)$, expressed as $W_c \in d_0 \times M$. Here $d_0$ is the output dimension of $f(x)$. $W_c$ is learned to maximize the similarity between CBL's output $f_c(x) = W_c f(x)$ and CLIP's activation matrix $P$. This incentivizes the $k$-th neuron to have an

activation pattern similar to CLIP with the $k$-th concept. We denote $k$-th neuron's, activation pattern as $q_k^\top = [f_{c,k}(x_1), ..., f_{c,k}(x_N)]^\top$. $W_c$ is then optimized by minimizing the following objective: $\sum_{i=1}^{M} -\frac{\bar{q}_i^3 \cdot \bar{P}_{:,i}^3}{\|\bar{q}_i^3\|_2 \|\bar{P}_{:,i}^3\|_2}$ where $\bar{q}$ means the vector is normalized to have mean 0 and standard derivation 1. Finally, LF-CBM learns a sparse linear prediction layer with weight $W_F$ and bias $b_F$. The optimization goal is in Eq. (1):

$$\min_{W_F, b_F} \|W_F f_c(\mathbf{X}_{\text{train}}) + b_F - y\|_2^2 + \lambda R_\alpha(W_F) \tag{1}$$

where $R_\alpha = 0.5(1-\alpha)\|W_F\|_2^2 + \alpha\|W_F\|_1$. Compared with LF-CBM, our CC-CBM is tailored for continual learning with a new learning procedure that enables it to learn a series of tasks, which generalizes CBM-based methods to continual learning. Recent work (Marconato et al., 2022) investigates CBM in continual learning setting. The main work of this paper is to find how concepts diverse or conserve under fine-tune strategy. However, it's not considering image classification accuracy in continual learning setting, which is different than our goal. Meanwhile, their strategy does not allow extension of concept set, which limits the ability in continual learning.

## 3 CONCEPT CONTROLLER - AN INTERPRETABLE REGULARIZATION-BASED METHOD FOR CONTINUAL LEARNING

**Overview.** To mitigate catastrophic forgetting via interpretable methods, our objective is to control human-understandable concepts that models learn from tasks. We propose two frameworks: the **Concept Controller** (CC) in this section and another new approach, the **Concept Controller CBM** (CC-CBM) in section 4. The CC framework first identity the interpretable neurons that a model has learned from previous tasks, then it freezes these neurons to prevent forgetting and reuse them in new tasks. Building upon this idea, CC-CBM aims to systematically insert and freeze interpretable neurons relevant to tasks.

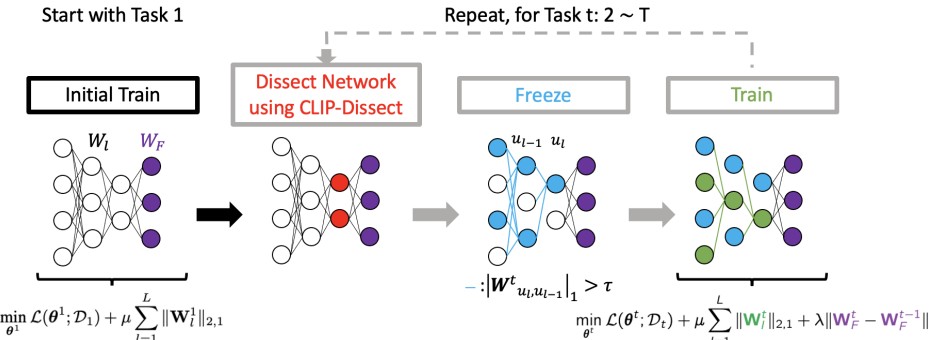

Figure 2: **Concept Controller**'s procedure.

**Key Idea.** CC is presented in Figure 2, and works in 4 steps described in the following paragraphs. To start with, let us introduce the notations: Convolutional neural networks can be divided into two parts - the convolutional part and the prediction part. First, suppose the convolutional part $\{\mathbf{W}_l\}_{l=1}^{L}$ be $L$ layers. For layer $l$, it has $U_l$ filters, and the parameter tensor is denoted as $\mathbf{W}_l$. Second, suppose the prediction part $\mathbf{W}_F$ has the Avgpool layer and the fully connected layers. The whole model parameter $\boldsymbol{\theta}$ is defined as $\boldsymbol{\theta} = \{\mathbf{W}_l\}_{l=1}^{L} \cup \mathbf{W}_F$.

**Step 1: Initial Training.** When training on the first task ($t = 1$), we try to promote the sparsity of the neural network. For the convolutional layers, we train the neural network with matrix $\ell_{2,1}$ norm regularization on its filters, which decreases the number of effective filters. Formally, the training goal is to optimize Eq. (2):

$$\min_{\boldsymbol{\theta}^1} \mathcal{L}(\boldsymbol{\theta}^1; \mathcal{D}_1) + \mu \sum_{l=1}^{L} \|\mathbf{W}_l^1\|_{2,1} \tag{2}$$

where $\mathcal{L}$ is cross entropy loss function and $\mu$ is the hyperparameter of the regularization term. We use superscript on the model parameter to denote task number.

**Step 2: Dissecting Network.** For the training process on task $t$, $t > 1$, we first use CLIP-Dissect (Oikarinen & Weng, 2022) on the last convolutional layer. CLIP-Dissect returns $n_t$ concept units $\bar{U}^t = \{\bar{u}_i^t\}_{i=1}^{n_t}$ in the last convolutional layer. A unit is a "interpretable unit" with a specific concept when its similarity score exceeds the threshold $\eta$. The detail of hyperparameter $\eta$ and the accuracy analysis of interpretable units are in CLIP-Dissect (Oikarinen & Weng, 2022).

**Step 3: Freeze Interpretable Neurons.** Inspired by DEN (Yoon et al., 2017), we perform Breadth First Search (BFS) from the last convolutional layer to the input layer, aim to find out all units that are related to a concept unit. Specially, we define two units $u_l, u_{l-1}$ from layer $l$ and $l - 1$ are connected if the $\ell_1$ norm of the filter $\mathbf{W}_{u_l, u_{l-1}}^t$ exceeds threshold $\tau$, and the BFS result of a concept unit is denoted as a "subnetwork" (Yoon et al., 2017). We find and freeze the subnetworks of all interpretable concept units. We freeze the subnetworks by setting the gradient of the subnetwork's parameters to zero. We implement two ways to freeze the subnetworks:

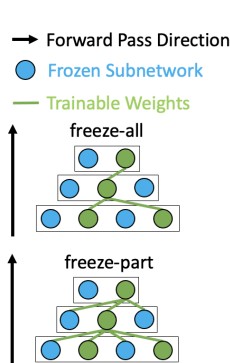

- **freeze-all** Freeze all weights connected to neurons in any concept units' subnetwork.

- **freeze-part** Only freeze incoming weights to all neurons in the subnetworks, leaving outgoing weights trainable.

An illustration of these two methods is in Figure 3. Note that the CC framework is light since it does not need extra memory storage to store previous tasks' samples. Previous tasks' knowledge is stored within the model. Meanwhile, CC is transparent because the important neurons are human-understandable.

Figure 3: The illustration of two freezing subnetwork methods. Green nodes and lines stand for trainable unit and weights.

**Step 4: Train.** Previous works (Lesort et al., 2021; Mirzadeh et al., 2022b) state that the Avgpool and the last layer are important for continual learning. Inspired by the EWC strategy, we train the neural network with regularization on the Avgpool and fully-connected layer. The regularization term aims to prevent parameter $\mathbf{W}_F$ from changing too much. Formally, the training goal is to optimize Eq. (3):

$$\min_{\boldsymbol{\theta}^t} \mathcal{L}(\boldsymbol{\theta}^t; \mathcal{D}_t) + \mu \sum_{l=1}^{L} \|\mathbf{W}_l^t\|_{2,1} + \lambda \|\mathbf{W}_F^t - \mathbf{W}_F^{t-1}\|_F \quad (3)$$

We will repeat step 2 to 4 for task 2, 3 ... until the last task $T$. CC's algorithm is summarized in the appendix A.4.

**Discussion on Interpretability.** CC uses CLIP-Dissect (Oikarinen & Weng, 2022) as the interpretability tool to dissect network and find interpretable units in the model. CLIP-Dissect provides very high concept accuracy for interpreting models as they studied in the Tables 2, 3 and 6 of the paper, which means it can interpret individual units well.

**Ability to Share Concepts Among Tasks.** CC is designed to keep the learned concepts from old tasks. Experiment results in Section 5.2 and Appendix A.7 prove our method's ability.

## 4   CONCEPT CONTROLLER CBM - AN INTERPRETABLE ARCHITECTURE-BASED METHOD FOR CONTINUAL LEARNING

In Section 3, we described how we can use **Concept Controller** to discover concept units in models without prior knowledge of the current task, and control these concept units to mitigate catastrophic forgetting. CC is the first interpretable approach in regularization-based category. In this section, we discuss how we can further leverage the task's information to control the concept units based on CC, which is a new approach in architecture-based category. We propose **Concept Controller CBM** (CC-CBM), which can transform any neural network backbone to a interpretable CBM for continual learning by combining CC with LF-CBM in section 2.2. CC-CBM is able to preserve concept units learned from previous tasks, and insert new concept units for new tasks.

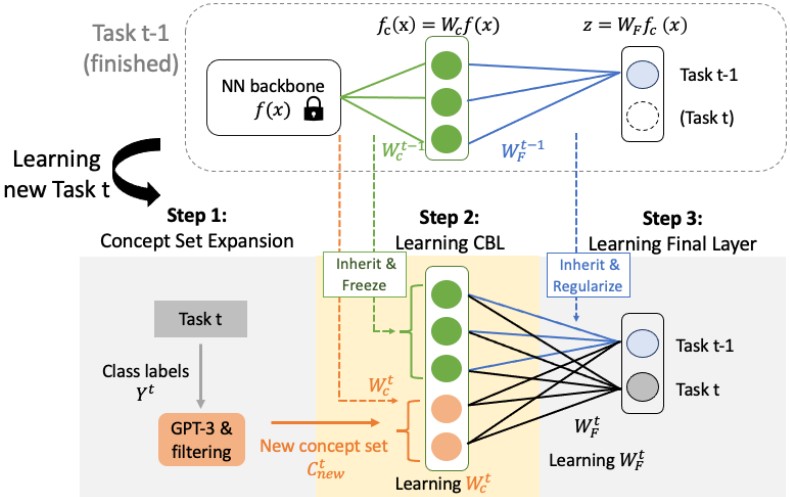

Figure 4: CC-CBM's procedure.

We illustrate CC-CBM's procedure in Figure 4. For the first task, the learning procedure is the same as training LF-CBM (section 2.2) on the first task. In the continual learning setting, after learning the task $t \in \{1, ..., T\}$, the CBM has concept set $\mathcal{C}^t$ with $M^t$ concepts, concept bottleneck layer (CBL) $W_c^t \in R^{M^t \times d_0}$ and the prediction layer $W_F^t \in R^{d_z \times M^t}, b_F^t \in R^{d_z}$ where $d_z$ is the number of output classes. The backbone $f(x)$ is frozen, and the CBL function is $f_c^t(x) = W_c^t f(x)$. Each task $t$ is associated with additional concept set $C_{new}^t$ with $N^t$ concepts which are generated using GPT-3 as described in (Oikarinen et al., 2023). Follow LF-CBM (Oikarinen et al., 2023) Section 3.1, we use several techniques to improve the quality of $C_{new}^t$. In the following sections for CC-CBM's learning procedure, we consider the case that the model has learned $t - 1$ tasks, and it is going to learn the new task $t$.

**Step 1: Concept Set Expansion.** Given the concept set from $t - 1$ tasks as $\mathcal{C}^{t-1}$, we form a new concept set $\mathcal{C}^t$ by adding all concepts in $C_{new}^t$ into $\mathcal{C}^{t-1}$. In the case some concepts in $C_{new}^t$ are already in $\mathcal{C}^{t-1}$, we still add them since two same concepts in different tasks might have different characteristics such as color and shape. For example, concept "ship" might refer to "vessel" or "cargo ship" in different tasks. After the expansion, there are $M^t = M^{t-1} + N^t$ concepts in $\mathcal{C}^t$.

**Step 2: Learning the Concept Bottleneck Layer (CBL).** To preserve previously learned concepts, we keep the weights of the existing concepts from the CBL of the previous task. Given a new CBL for learning the task $t$ is $W_c^t \in R^{M^t \times d_0}$, we first inherit the CBL's weights from the previous model. Specifically, we copy the previous concepts' weights in $W_c^{t-1} \in R^{M^{t-1} \times d_0}$ to the same concepts in $W_c^t$. Therefore, $M^{t-1}$ concepts in $W_c^t$ inherit the identical weights from $W_c^{t-1}$. The second step is to learn $W_c^t$ using the procedure of LF-CBM described in section 2.2 to get $\hat{W}_c^t$. The second step can be done in two ways:

1. **Finetune**: We learn $W_c^t$ on task $t$ without additional constraints. We only initialize $W_c^t$ with old concepts but these concepts will be changed on the new task.

2. **Concept Controller**: We freeze the previous concepts' corresponding weights in $W_c^t$ to prevent them from changing. This strategy enables us to learn the weights of new concepts without affecting the previously learned ones. This strategy makes CC-CBM light and interpretable. It eliminates the need for additional memory storage for previous tasks' samples, and provides transparency regarding the retained knowledge.

**Step 3: Learning the Prediction Layer.** First, we inherit concepts' weights in $W_F^{t-1}$ to $W_F^t$ before the learning process, and same for the $b_F^t$. The idea is similar to step 2, where $M^{t-1}$ concepts in $W_F^t$ inherit the same weights from $W_F^{t-1}$. Again, the **Finetune** strategy is to learn $W_F^t$ without any constraints. The optimization of $W_F^t$ is same as Eq. (1). On the other hand, the **Concept Controller** strategy for LF-CBM follows the similar idea as CC's step 4. We aim to learn the prediction layer

with regularization. The training goal is to optimize Eq. (4):

$$\min_{W_F^t, b_F^t} \mathcal{L}(W_F^t, b_F^t; f_c^t(\mathcal{X}^t), \mathcal{Y}^t) + \lambda R_\alpha(W_F^t) + \gamma(\|W_F^t - W_F^{t-1}\|_2^2 + \|b_F^t - b_F^{t-1}\|_2^2) \quad (4)$$

where $\mathcal{L}$ is a specific loss function and $\gamma$ is the hyperparameter of the regularization term.

**Discussion on Interpretability** LF-CBM (Oikarinen et al., 2023) provides CC-CBM interpretability. While we can't directly measure its concept accuracy in our setting, the concepts learned by LF-CBM were shown to be accurate in a crowdsourced evaluation (Oikarinen et al., 2023).

**Ability to Share Concepts Among Tasks** CC-CBM is designed to keep the learned concepts from old tasks. Experiment results in Section 5.2 and Appendix A.8 prove our method's ability.

## 5 EXPERIMENT

**Dataset and Experiment setup** To evaluate our methods, we perform experiments on three datasets: CIFAR-10, CIFAR-100 (Krizhevsky et al., 2009), and TinyImageNet (Le & Yang, 2015). CIFAR-10/ CIFAR-100 and TinyImageNet are standard image classification benchmarks. Both CIFAR-10 and CIFAR-100 have 50k training examples and 10000 testing examples with 10 classes and 100 classes respectively. TinyImageNet has 200 classes with 500 training examples and 50 testing examples per class. We consider $T = 5, 10, 20$ tasks, where each task consists of $\frac{100}{T}\%$ of classes and their corresponding samples. Our experiments focus on the class incremental setting, which means tasks don't share the class labels. We use ResNet18 (He et al., 2016) as our experiment model. We split each dataset by 3 different random seeds, and run each class distribution for 3 times.

**Evaluation Metrics** Following (Mirzadeh et al., 2022b; Chaudhry et al., 2018a), we use the standard evaluation metrics to evaluate our methods. Define $a_{i,j}$ as model's accuracy on $j$-th task after learning $i$-th task, $i \geq j$. When testing the performance on $t$-th task, the metrics' definitions are as follows:

- **Average Accuracy** ($A_t = \frac{1}{t}\sum_{i=1}^{t} a_{t,i}$): Measures the average model performance.
- **Average Forgetting** ($F_t = \frac{1}{t-1}\sum_{i=1}^{t-1} \max_{j \in (1,...,t-1)}(a_{j,i} - a_{t,i})$): Measures model performance drop on previous tasks.

However, these standard metrics only reflect models' performance at the final stage. Following recent works (Zhu et al., 2022; Carta et al., 2023; Zhou et al., 2023; Caccia et al., 2020; 2021; Koh et al., 2021), we also evaluation models throughout the stream. Specifically, we also report:

- **Average Incremental Accuracy** ($\bar{A}_T = \frac{1}{T}\sum_{t=2}^{T} A_t$)
- **Average Incremental Forgetting** ($\bar{F}_T = \frac{1}{T}\sum_{t=2}^{T} F_t$)

### 5.1 QUANTITATIVE COMPARISON RESULTS

We perform experiments on the following continual learning baselines:

- Finetune: the standard method where models are updated continuously on a series of tasks
- Exemplar-free methods
    - Category **(i)**: EWC (Kirkpatrick et al., 2017), SI (Zenke et al., 2017), LwF (Li & Hoiem, 2017)
    - Category **(ii)**: Adam-NSCL (Wang et al., 2021), SSRE (Zhu et al., 2022) in Appendix A.12
- Exemplar-based methods
    - Category **(iii)**: GEM (Lopez-Paz & Ranzato, 2017), MIR (Aljundi et al., 2019), DER (Buzzega et al., 2020)

For CC, "CC-freeze-x" ($x \in \{all,part\}$) means **Concept Controller** with implementation freeze-all/ freeze-part in step 3. "CC-freeze-x-GEM" and "CC-freeze-x-MIR" means combining "CC-freeze-x" with GEM and MIR respectively. For CBM based methods, "Finetune-CBM" means using **Finetune** strategy in step 2 and step 3. "CC-CBM" means using **Concept Controller** strategy in step 2 and step 3. Similarly, "CC-CBM-GEM and "CC-CBM-MIR" means combining "CC-CBM" with GEM

and MIR respectively. For baseline strategies, we use the implementations from a continual learning library Avalanche (Lomonaco et al., 2021). We also use Avalanche to implement our methods.

Here we discuss $T = 5$ experiment results in $\{\bar{A}_T, \bar{F}_T\}$. For experiment results of $\{T = 10, 20\}$, different metrics $\{A_T, F_T\}$ and comparison with SSRE (Zhu et al., 2022), please see Appendix A.10, A.9, and A.12 respectively. For CC, the accuracy comparisons with existing works are in Table 1. Compared with the exemplar-free methods, our method outperforms existing works by up to 1.4% in $\bar{A}_T$ and up to 1.4% in $\bar{F}_T$. Most of the time CC performs better using freeze-all 3 than with freeze-part. Meanwhile, the performance is comparable to or even better than exemplar-based methods in some benchmarks. When combining CC with exemplar-based methods, both freeze-all and freeze-part make exemplar-based methods have better accuracies in balanced average accuracy $\bar{A}_T$, and have lower balanced average forgetting $\bar{F}_T$. The experiment results show that CC can make models more effective and forget less, either working independently or combining with other methods.

Table 1: Accuracy comparison for CC. $\uparrow$ means larger values are better, while $\downarrow$ means smaller values are better. The **Improvement** is compared with the strongest baseline for each block. Our methods outperform the baselines on both $\bar{A}_T$ and $\bar{F}_T$ by large margins.

| | CIFAR-10, 5T | | CIFAR-100, 5T | | TinyImagenet, 5T | |
|---|---|---|---|---|---|---|
| | $\bar{A}_T \uparrow$ | $\bar{F}_T \downarrow$ | $\bar{A}_T \uparrow$ | $\bar{F}_T \downarrow$ | $\bar{A}_T \uparrow$ | $\bar{F}_T \downarrow$ |
| **Baseline in Category (i)** | | | | | | |
| Finetune | $27.46 \pm 0.89$ | $95.86 \pm 1.28$ | $20.47 \pm 0.67$ | $63.02 \pm 0.45$ | $16.82 \pm 2.18$ | $49.80 \pm 0.57$ |
| EWC | $30.05 \pm 0.79$ | $94.13 \pm 2.26$ | $20.97 \pm 0.55$ | $62.56 \pm 0.29$ | $16.19 \pm 2.65$ | $48.42 \pm 0.39$ |
| SI | $29.64 \pm 0.35$ | $94.21 \pm 1.66$ | $17.75 \pm 1.37$ | $59.31 \pm 1.93$ | $13.07 \pm 2.57$ | $44.71 \pm 2.28$ |
| LwF | $30.16 \pm 0.23$ | $94.94 \pm 1.33$ | $12.74 \pm 2.15$ | $63.66 \pm 2.40$ | $16.09 \pm 3.25$ | $49.43 \pm 1.26$ |
| **Baseline in Category (ii)** | | | | | | |
| Adam-NSCL | $30.23 \pm 1.02$ | $94.82 \pm 0.53$ | $17.45 \pm 2.35$ | $59.54 \pm 3.04$ | $17.90 \pm 2.57$ | $44.98 \pm 0.74$ |
| **Ours** | | | | | | |
| CC-freeze-all | $\mathbf{31.55 \pm 0.13}$ | $\mathbf{92.69 \pm 0.81}$ | $\mathbf{22.37 \pm 1.20}$ | $\mathbf{58.75 \pm 0.26}$ | $\mathbf{18.19 \pm 0.76}$ | $\mathbf{43.39 \pm 0.92}$ |
| CC-freeze-part | $30.55 \pm 0.84$ | $94.47 \pm 1.12$ | $21.73 \pm 0.79$ | $60.51 \pm 0.35$ | $18.08 \pm 0.56$ | $46.00 \pm 0.66$ |
| **Improvement** | **1.32** | **1.44** | **1.40** | **0.56** | **0.29** | **1.32** |
| **Baseline in Category (iii)** | | | | | | |
| GEM | $34.59 \pm 0.05$ | $90.00 \pm 3.80$ | $23.02 \pm 1.65$ | $60.63 \pm 4.12$ | $11.29 \pm 2.62$ | $41.30 \pm 1.67$ |
| MIR | $28.97 \pm 2.34$ | $89.80 \pm 0.99$ | $27.26 \pm 0.93$ | $53.02 \pm 2.84$ | $17.81 \pm 0.82$ | $44.60 \pm 2.43$ |
| DER | $32.39 \pm 1.76$ | $\mathbf{72.24 \pm 1.32}$ | $26.49 \pm 1.64$ | $54.06 \pm 1.41$ | $15.27 \pm 0.89$ | $48.01 \pm 0.06$ |
| **Ours** | | | | | | |
| CC-freeze-all-GEM | $35.60 \pm 0.62$ | $94.93 \pm 0.66$ | $\mathbf{28.18 \pm 2.36}$ | $\mathbf{42.61 \pm 2.12}$ | $12.49 \pm 1.87$ | $\mathbf{37.78 \pm 1.04}$ |
| CC-freeze-part-GEM | $\mathbf{37.00 \pm 0.96}$ | $92.22 \pm 2.00$ | $25.62 \pm 2.08$ | $52.39 \pm 3.24$ | $12.06 \pm 1.58$ | $42.67 \pm 2.79$ |
| CC-freeze-all-MIR | $30.23 \pm 1.54$ | $91.04 \pm 2.57$ | $27.04 \pm 1.08$ | $51.66 \pm 3.11$ | $19.95 \pm 2.41$ | $43.08 \pm 0.95$ |
| CC-freeze-part-MIR | $30.75 \pm 1.37$ | $91.07 \pm 2.04$ | $27.59 \pm 0.62$ | $52.77 \pm 2.75$ | $\mathbf{20.02 \pm 2.58}$ | $43.87 \pm 2.25$ |
| **Improvement** | **2.41** | -18.80 | **0.92** | **10.41** | **2.21** | **3.52** |

Since the backbone of our CBM is pre-trained on Places365 dataset (Zhou et al., 2017), we also pre-trained existing methods' models on Places365 dataset for a fair comparison. The accuracy comparisons for CBM-based methods are in Table 2. Due to the architecture of Label-free CBM 2.2, the Finetune-CBM outperforms existing exemplar-free methods in balanced average forgetting $\bar{F}_T$. CC-CBM even has better performance on both metrics. Compared with CC, the performance of CC-CBM is better since it's $\bar{F}_T$ outperforms baselines by up to 9.1%. Meanwhile, combining CC-CBM with exemplar-based methods yields improved performance in both metrics for the exemplar-based methods. We also did ablation studies in Appendix A.5 to understand the impact of components in CC and CC-CBM.

## 5.2 DISCUSSION ON CONCEPT EVOLUTION

In addition to analyzing the prediction accuracy of our methods, we also examine the learned concepts of our methods by leveraging their interpretability. For CC, we study the evolution of the concepts represented by neurons as we train across different tasks. We analyze the case which we group similar classes into the same task, so we can recognize which task a concept belongs to easier. For CIFAR-100 and TinyImagenet, the class distributions are in appendix A.15. We try to maximize the diversity between tasks, which makes tasks share less concepts. First, we use CLIP-Dissect (Oikarinen & Weng, 2022) to analyse how many units are still detecting the same concept after

Table 2: Accuracy comparison for CC-CBM. All models are pre-trained on the Place365 dataset Zhou et al. (2017). ↑ means larger values are better, while ↓ means smaller values are better. The **Improvement** is compared with the strongest baseline for each block. Our methods clearly outperform the baselines on both $\bar{A}_T$ and $\bar{F}_T$.

| | CIFAR-10, 5T | | CIFAR-100, 5T | | TinyImagenet, 5T | |
|---|---|---|---|---|---|---|
| | $\bar{A}_T \uparrow$ | $\bar{F}_T \downarrow$ | $\bar{A}_T \uparrow$ | $\bar{F}_T \downarrow$ | $\bar{A}_T \uparrow$ | $\bar{F}_T \downarrow$ |
| **Baseline in Category (i)** | | | | | | |
| Finetune | $29.13 \pm 0.28$ | $97.78 \pm 0.78$ | $22.68 \pm 0.88$ | $75.69 \pm 2.25$ | $20.50 \pm 0.45$ | $66.52 \pm 2.10$ |
| EWC | $30.79 \pm 0.21$ | $97.92 \pm 0.78$ | $22.20 \pm 1.56$ | $74.93 \pm 2.34$ | $19.68 \pm 1.82$ | $65.26 \pm 2.54$ |
| SI | $30.25 \pm 0.72$ | $96.55 \pm 1.70$ | $22.70 \pm 1.20$ | $74.44 \pm 2.45$ | $18.87 \pm 1.71$ | $62.44 \pm 0.68$ |
| LwF | $30.76 \pm 0.31$ | $97.75 \pm 0.84$ | $24.20 \pm 2.29$ | $74.41 \pm 3.33$ | $20.48 \pm 2.36$ | $64.17 \pm 3.08$ |
| **Baseline in Category (ii)** | | | | | | |
| Adam-NSCL | $30.78 \pm 0.82$ | $96.82 \pm 2.35$ | $23.37 \pm 2.14$ | $53.87 \pm 3.19$ | $20.17 \pm 2.04$ | $58.28 \pm 3.02$ |
| **Ours** | | | | | | |
| Finetune-CBM | $30.39 \pm 0.67$ | $93.09 \pm 2.22$ | $\mathbf{24.99 \pm 0.79}$ | $59.29 \pm 1.40$ | $21.13 \pm 1.66$ | $61.57 \pm 1.42$ |
| CC-CBM | $\mathbf{32.25 \pm 0.76}$ | $\mathbf{88.58 \pm 0.30}$ | $24.25 \pm 0.86$ | $\mathbf{47.62 \pm 1.52}$ | $\mathbf{21.30 \pm 1.55}$ | $\mathbf{49.11 \pm 1.74}$ |
| **Improvement** | **1.46** | **7.97** | **0.79** | **6.25** | **0.80** | **9.17** |
| **Baseline in Category (iii)** | | | | | | |
| GEM | $36.27 \pm 1.86$ | $69.38 \pm 2.88$ | $24.27 \pm 1.57$ | $74.28 \pm 2.54$ | $10.54 \pm 0.73$ | $41.49 \pm 1.62$ |
| MIR | $33.20 \pm 2.92$ | $75.51 \pm 1.94$ | $24.32 \pm 2.09$ | $61.22 \pm 1.69$ | $11.58 \pm 2.08$ | $44.17 \pm 2.06$ |
| DER | $33.77 \pm 1.28$ | $76.75 \pm 1.24$ | $24.21 \pm 2.66$ | $68.11 \pm 2.15$ | $\mathbf{19.18 \pm 0.78}$ | $57.51 \pm 3.96$ |
| **Ours** | | | | | | |
| CC-CBM-GEM | $36.37 \pm 1.74$ | $\mathbf{44.72 \pm 2.71}$ | $25.41 \pm 2.06$ | $66.91 \pm 2.16$ | $12.14 \pm 0.53$ | $\mathbf{38.23 \pm 0.42}$ |
| CC-CBM-MIR | $\mathbf{36.61 \pm 1.77}$ | $70.45 \pm 1.89$ | $\mathbf{31.03 \pm 2.23}$ | $\mathbf{60.39 \pm 1.06}$ | $14.86 \pm 1.10$ | $46.24 \pm 2.81$ |
| **Improvement** | **0.34** | **24.66** | **6.71** | **0.83** | -4.32 | **3.26** |

learning a new task. The results are in Table 3. Compared with existing methods, our method has better ability to retain knowledge of concepts learned from previous tasks. We also do a case study to understand how well our method preserves the concept units from the previous tasks. Table 8 shows results for some example neurons. Some concepts are preserved after learning unrelated new tasks, which gives us insight on how our method helps avoid catastrophic forgetting.

For CC-CBM, we also studied the evolution of concepts across different tasks. Figure 5 shows an example for CIFAR-100 under 5-tasks scenario, studies on other datasets is in appendix A.8. Compared with Finetune-CBM, we can see CC-CBM is much better at retaining and using knowledge of concepts learned from previous tasks, which help us understand how our method against catastrophic forgetting.

Table 3: The ratio of units which still detect the same concepts they detected in the last task. Tasks' classes are in appendix A.15. Our methods outperform existing works for preserving concepts.

| Method | CIFAR100, 5T | | | | TinyImagenet, 5T | | | |
|---|---|---|---|---|---|---|---|---|
| | Task 2 | Task 3 | Task 4 | Task 5 | Task 2 | Task 3 | Task 4 | Task 5 |
| **Baseline** | | | | | | | | |
| Finetune | 0 | 0 | 0 | 0 | 0 | 0 | 0 | 0.086 |
| EWC | 0 | 0 | 0 | 0 | 0 | 0.030 | 0.037 | 0 |
| SI | 0 | 0 | 0 | 0 | 0.028 | 0.191 | 0.200 | 0.250 |
| LwF | 0.019 | 0.011 | 0.176 | 0 | 0.114 | 0.191 | 0.393 | 0.365 |
| GEM | 0 | 0.125 | 0.076 | 0.153 | 0 | 0 | 0 | 0 |
| MIR | 0 | 0 | 0 | 0.250 | 0.028 | 0 | 0 | 0 |
| **Ours** | | | | | | | | |
| CC-freeze-all | **0.500** | **0.500** | 0.571 | **0.700** | **0.739** | **0.771** | **0.812** | **0.812** |
| CC-freeze-part | 0.125 | 0.133 | **0.727** | 0.695 | 0.515 | 0.488 | 0.544 | 0.666 |

# 6 CONCLUSION

In this work, we have presented **Concept Controller**, a high performance, light and interpretable framework to mitigate catastrophic forgetting problem in continual learning. We have shown **Concept Controller** and its extended version (CC-CBM) outperforms previous continual learning methods by up to 1.4% in average incremental accuracy, and it can further improve them by up to 24% in average incremental forgetting. It can preserve learned knowledge in an effective and interpretable way.

## REPRODUCIBILITY STATEMENT

We describe training details and hyperparameters in Appendix A.1. We also fix the task distribution of 5-tasks scenario to reproduce our methods' interpretability results. Please see Appendix A.15 for the details. The code and full training details will be released to public upon acceptance.

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

# A APPENDIX

In this section we provide a brief overview of the Appendix contents.

- A.1: Training and Computing Details
- A.2: Broader Impacts
- A.3: Previous Continual Learning Methods
- A.4: **Concept Controller** Algorithm
- A.5: Ablation Study
- A.6: Ablation on CC-CBM Sparsity
- A.7: CC's Concept Evolution
- A.8: Discussion of Concept Evolution for CC-CBM
- A.9: 5 task Standard Metric Results
- A.10: 10, 20 tasks Experiment Results
- A.11: Computational efficiency
- A.12: Comparison with SSRE
- A.15: Class Distribution

## A.1 TRAINING AND COMPUTING DETAILS

All models are trained on single NVIDIA V100s (32 GB SMX2). The hyperparameters used for our methods are in Table 4. The code and full training details will be released to public upon acceptance. For a fair comparison, the $\lambda$ and $\gamma$ in Eq. 3 and 4 are same as the hyperparameters in regularization term of EWC (Kirkpatrick et al., 2017), SI (Zenke et al., 2017) and LwF (Li & Hoiem, 2017). For GEM (Lopez-Paz & Ranzato, 2017) MIR (Aljundi et al., 2019) and DER (Buzzega et al., 2020), we experimented with the following memory buffer sizes per task: [20, 150, 500] and report results with 150 which has best average accuracy in 5-tasks scenario.

Table 4: Hyperparameters for our methods.

|  | CIFAR-10/ CIFAR-100 | TinyImageNet |
| --- | --- | --- |
| $\mu$ in Eq. 3 | $10^{-6}$ | $10^{-6}$ |
| $\lambda$ in Eq. 3 | 0.4 | 0.4 |
| $\tau$ in section 4 | 0.2 | 0.35 |
| $\lambda$ in Eq. 4 | 0 | 0 |
| $\gamma$ in Eq. 4 | 0.4 | 0.4 |

## A.2 BROADER IMPACTS

The improvement of mitigating forgetting by controlling concepts in models might have some potential negative impact in terms of privacy. For example, if an adversary only has access to model checkpoints but not model's training data, they can analyse the concept units in the model. This will give the attacker some information about how the model was trained, and may allow them to extract some private information from the models without access to training data.

## A.3 PREVIOUS CONTINUAL LEARNING METHODS

The representative method in regularization-based methods category **(i)** is **Elastic Weight Consolidation (Kirkpatrick et al., 2017)**. Elastic Weight Consolidation (EWC) is a regularized-based method. The loss function in this strategy $L(\boldsymbol{\theta}^t) = L_{\text{CE}}(\boldsymbol{\theta}^t) + \frac{\lambda}{2} \sum_i F_i^t (\theta_i^t - \theta_i^{t-1})^2$ has a quadratic penalty term which is related to the difference between the parameters of the old task and the new task. Meanwhile, this penalty term is proportional to the diagonal of the Fisher information matrix F.

$\lambda$ is a hyperparameter, and $\theta^t, \theta^{t-1}$ stand for model parameters before and after training on a new task $t$ respectively.

Similarly, Zenke et al. (2017) adds a quadratic penalty term in the loss function, while estimating the importance of parameters during training. Li & Hoiem (2017) trains models with a knowledge distillation loss for old tasks and a regularization loss to outperform joint training.

The classic method in architecture-based methods category **(ii)** is **Dynamic Expandable Network (DEN) (Yoon et al., 2017)**. DEN focuses on the neural network's architecture aspect in the continual learning setting. This method has three steps. First, they train the network to be sparse on task $t - 1$. The loss function $\mathcal{L}(W_{1:L}^{t-1}; \mathcal{D}_{t-1}) + \mu \sum_{l=1}^{L} \|W_l^{t-1}\|_1$ has $\ell_1$-regularization of each layer's parameter $W_l^{t-1}$ which boosts the network's sparsity. Next, when training on the new task $t$, they do a breadth-first search to find neurons that are connected to output $o_t$, and the authors describe the search result as a "subnetwork". Therefore, they only need to train a subnetwork instead of the whole network, which is called *selective retraining*. This procedure can prevent parameters related to the previous task from being changed. If the performance is bad, which means the loss function exceeds the predefined threshold, they will expand the network's layers with some neurons and prune unnecessary ones. Finally, they measure the $l_2$-distance of the hidden neurons before and after training on a new task. If the distance exceeds the predefined threshold $\rho$, they will duplicate them and train the model again.

The classic method in replay-based methods category **(iii)** is **Gradient Episodic Memory (Lopez-Paz & Ranzato, 2017)**. Gradient Episodic Memory (GEM) is a replay-based method. It stores a subset of the observed examples from previous tasks, which is described as episodic memory. When training on new tasks, it regularizes the projection of the estimated gradient descent $g$ on the gradient descent of episodic memory $g_k$. The optimization goal is formalized as $\langle g, g_k \rangle \geq 0, \forall k < t$ when training on the task $t$. This regularization prevents the loss of episodic memory from increasing. (Rebuffi et al., 2017) stores a subset of samples for each class and uses the nearest-mean classifier on the data representation space.

## A.4 CONCEPT CONTROLLER ALGORITHM

The algorithm 1 summarizes the procedure of **Concept Controller**.

---

**Algorithm 1 Concept Controller**: Freeze the subnetworks of the concept units

---

**Require:** Dataset $\mathbb{D}$; regularization coefficient $\mu$; connection threshold $\tau$; regularization factor $\lambda$;
    Neural network parameters $\theta$

1: **for** $t \leftarrow 1,...,$T **do**
2:      **if** $t$ is 1 **then**
3:          Train $\theta^1$ on $D_t$ by solving Eq. 2
4:      **else**
5:          Train $\theta^t$ on $D_t$ by solving Eq. 3
6:      `ConceptUnit` $\leftarrow$ CLIP-Dissect($W^t$)
7:      `Prev-active` $\leftarrow$ `ConceptUnit`
8:      **for** layer $l \leftarrow$ L,...,1 **do**               ▷ Find the subnetwork of the concept units
9:          **for** Unit $u_l \leftarrow 1,...,U_l$ **do**
10:              **if** `Prev-active`$[u_l]$ is **True then**          ▷ $u_l$ is in subnetwork
11:                  **for** Unit $u_{l-1} \leftarrow 1,...,U_{l-1}$ **do**
12:                      **if** $\|\mathbf{W}_{u_l, u_{l-1}}^t\|_1 > \tau$ **then**        ▷ weight exceeds threshold
13:                          `Active`$[u_{l-1}] \leftarrow$ **True**
14:          **if** Using freeze-all **then**
15:              Freeze $\mathbf{W}_{u_l,:,:}^t, \forall$ `Prev-active`$[u_l]$ is **True**
16:              Freeze $\mathbf{W}_{:, u_{l-1}}^t, \forall$ `Active`$[u_{l-1}]$ is **True**
17:          **else if** Using freeze-part **then**
18:              Freeze $\mathbf{W}_{u_l,:,:}^t, \forall$ `Prev-active`$[u_l]$ is **True**
19:          `Prev-active` $\leftarrow$ `Active`

---

A.5 ABLATION STUDY

Besides comparing the results with existing methods, we perform ablation study to analyze the impact of the key components in our methods. In Table 5, we compare CC with (i) not freezing subnetwork in step 3, which is denoted as "CC w/o freeze", and (ii) not regularizing parameter $\mathbf{W}_F$ in step 4, which is denoted as "CC-freeze-all w/o reg" or "CC-freeze-part w/o reg". The experiment results show that combining two of them results in less forgetting and better average accuracy. We believe it's because they are necessary to each other. If the concept units' subnetworks are not frozen, the concepts might change and the regularization term may not work. On the other hand, if the $\mathbf{W}_F$ and $\mathbf{b_F}$ change when learning new tasks, then the concept units learned from previous tasks may not contribute to the final prediction.

For CC-CBM, we do a similar ablation study with (i) not freezing previously learned concepts when learning CBL in step 2, which is denoted as "CC-CBM w/o freeze", and (ii) not regularizing parameter $\mathbf{W}_F$ and $\mathbf{b}_F$ in step 3, which is denoted as "CC-CBM w/o reg". The result Table 6 shows a similar result. In conclusion, these ablation studies demonstrate the importance of both freezing subnetworks and regularizing the $\mathbf{W}_F$ and $\mathbf{b}_F$ to achieving better performance and mitigating forgetting.

Table 5: Experiment results of CC's ablation study. "CC w/o freeze": not freezing subnetwork. "CC-freeze-all w/o reg" and "CC-freeze-part w/o reg": not regularizing parameter $\mathbf{W}_F$ and $\mathbf{b}_F$. $\uparrow$ means larger values are better, while $\downarrow$ means smaller values are better. Preserving two components turns out have the best performances.

| | CIFAR-10, 5T | | CIFAR-100, 5T | | TinyImagenet, 5T | |
| --- | --- | --- | --- | --- | --- | --- |
| | $\bar{A}_T \uparrow$ | $\bar{F}_T \downarrow$ | $\bar{A}_T \uparrow$ | $\bar{F}_T \downarrow$ | $\bar{A}_T \uparrow$ | $\bar{F}_T \downarrow$ |
| CC w/o freeze | $29.77 \pm 0.12$ | $94.76 \pm 1.48$ | $19.27 \pm 2.53$ | $60.23 \pm 6.53$ | $15.51 \pm 2.14$ | $48.26 \pm 4.57$ |
| CC-freeze-all w/o reg | $30.13 \pm 0.62$ | $95.50 \pm 1.27$ | $19.32 \pm 2.02$ | $59.77 \pm 4.99$ | $14.56 \pm 2.37$ | $50.69 \pm 3.88$ |
| CC-freeze-part w/o reg | $30.17 \pm 0.58$ | $95.50 \pm 1.27$ | $21.21 \pm 1.65$ | $62.04 \pm 4.58$ | $15.07 \pm 2.50$ | $51.28 \pm 4.22$ |
| CC-freeze-all | $\mathbf{31.55 \pm 0.13}$ | $\mathbf{92.69 \pm 0.81}$ | $\mathbf{22.37 \pm 1.20}$ | $\mathbf{58.75 \pm 0.26}$ | $\mathbf{18.19 \pm 0.76}$ | $\mathbf{43.39 \pm 0.92}$ |
| CC-freeze-part | $30.55 \pm 0.84$ | $94.47 \pm 1.12$ | $21.73 \pm 0.79$ | $60.51 \pm 0.35$ | $18.08 \pm 0.56$ | $46.00 \pm 0.66$ |

Table 6: Experiment results of CC-CBM's ablation study. "CC-CBM w/o freeze": not freezing previous concepts. "CC-CBM w/o reg": not regularizing parameter $\mathbf{W}_F$ and $\mathbf{b}_F$. Preserving both has the best performance.

| | CIFAR-10, 5T | | CIFAR-100, 5T | | TinyImagenet, 5T | |
| --- | --- | --- | --- | --- | --- | --- |
| | $\bar{A}_T \uparrow$ | $\bar{F}_T \downarrow$ | $\bar{A}_T \uparrow$ | $\bar{F}_T \downarrow$ | $\bar{A}_T \uparrow$ | $\bar{F}_T \downarrow$ |
| CC-CBM w/o freeze | $29.78 \pm 2.45$ | $\mathbf{87.76 \pm 1.07}$ | $18.01 \pm 3.42$ | $49.50 \pm 2.58$ | $17.30 \pm 2.89$ | $51.64 \pm 4.00$ |
| CC-CBM w/o reg | $28.75 \pm 1.62$ | $91.10 \pm 2.20$ | $20.42 \pm 2.68$ | $58.72 \pm 1.04$ | $19.76 \pm 2.26$ | $62.24 \pm 2.85$ |
| CC-CBM | $\mathbf{32.25 \pm 0.76}$ | $88.58 \pm 0.30$ | $\mathbf{24.25 \pm 0.86}$ | $\mathbf{47.62 \pm 1.52}$ | $\mathbf{21.30 \pm 1.55}$ | $\mathbf{49.11 \pm 1.74}$ |

### A.6 ABLATION ON CC-CBM SPARSITY

In this section we experimented with some modifications to $W_F$ in our CBM and report their results. In our main results for CC-CBM and Finetune-CBM, we used a dense final layer $W_F$ instead of the sparse final layer used by (Oikarinen et al., 2023) as we found that to have best performance, and we are less focused on interpretable final decisions. Below we compare the results for CBM between dense $W_F$, and sparse $W_F$ (-S). For the sparse $W_F$, we used $\lambda = 10^{-6}$ in Eq. 4. The experiment results are in Table 7. We found that sometimes sometimes sparse $W_F$ makes models forget less. However, dense $W_F$ had the best performance in general.

Table 7: Experiment results of CBM based methods on 3 datasets. "S": sparse $W_F$. Dense $W_F$ results in best performance in average.

|  | CIFAR-10,5T | | CIFAR-100, 5T | | TinyImageNet, 5T | |
| --- | --- | --- | --- | --- | --- | --- |
|  | $\bar{A}_T \uparrow$ | $\bar{F}_T \downarrow$ | $\bar{A}_T \uparrow$ | $\bar{F}_T \downarrow$ | $\bar{A}_T \uparrow$ | $\bar{F}_T \downarrow$ |
| Finetune-CBM | $30.39 \pm 0.67$ | $93.09 \pm 2.22$ | $\mathbf{24.99 \pm 0.79}$ | $59.29 \pm 1.40$ | $21.13 \pm 1.66$ | $61.57 \pm 1.42$ |
| CC-CBM | $\mathbf{32.25 \pm 0.76}$ | $88.58 \pm 0.30$ | $24.25 \pm 0.86$ | $47.62 \pm 1.52$ | $\mathbf{21.30 \pm 1.55}$ | $49.11 \pm 1.74$ |
| Finetune-CBM-S | $28.84 \pm 1.35$ | $91.74 \pm 2.22$ | $20.23 \pm 2.79$ | $58.29 \pm 3.12$ | $19.50 \pm 2.33$ | $61.57 \pm 2.87$ |
| CC-CBM-S | $30.11 \pm 3.04$ | $\mathbf{87.58 \pm 0.30}$ | $19.20 \pm 3.77$ | $\mathbf{46.62 \pm 2.52}$ | $19.50 \pm 3.03$ | $\mathbf{46.11 \pm 3.48}$ |
| CC-CBM-GEM | $\mathbf{36.37 \pm 1.74}$ | $\mathbf{44.72 \pm 2.71}$ | $\mathbf{25.41 \pm 2.06}$ | $66.91 \pm 2.16$ | $\mathbf{12.14 \pm 0.53}$ | $38.23 \pm 0.42$ |
| CC-CBM-GEM-S | $36.32 \pm 2.90$ | $45.28 \pm 4.17$ | $25.40 \pm 4.13$ | $\mathbf{65.21 \pm 3.28}$ | $5.32 \pm 1.06$ | $39.23 \pm 0.85$ |

### A.7 CC'S CONCEPT EVOLUTION

Table 3 and 8 shows the CC's concept evolution when grouping similar classes together. The analysis is in Section 5.2.

Table 8: Concept evolution for CC in freeze-all implementation. We analyse the concept evolution in the layer 4 of ResNet18. The blue concept means the concept is related to the current task, while the green concept means it is unrelated. "x" stands for non-interpretable units. The results show that CC can preserve the concepts from previous classes while learning unrelated tasks.

| TinyImagenet, 5T | | | | | |
| --- | --- | --- | --- | --- | --- |
|  | task 1 | task 2 | task 3 | task 4 | task 5 |
| Classes | Big objects | Human-made small objects | Big animals & Natural scenes | Small animals & Sea animals | Food & Clothes & Others |
| Unit 85 | x | Kitchen | Kitchen | Kitchen | Kitchen |
| Unit 123 | x | Bedroom | Bedroom | Bedroom | Bedroom |
| Unit 129 | Bus | Bus | Bus | Bus | Bus |
| CIFAR-100, 5T | | | | | |
|  | task 1 | task 2 | task 3 | task 4 | task 5 |
| Classes | Small animals & Sea animals | Natural scenes & Plants | Big animals | Big objects | Others |
| Unit 307 | x | Kitchen | Kitchen | x | x |
| Unit 310 | x | x | x | Highway | Highway |

Besides grouping similar classes into the same task, we also analyse the cases where class labels are distributed randomly. Following the same procedure as in section 5.2, we analyse how many units are still detecting the same concept after learning a new task. The average results for three datasets are in Table 9 and 10. Similar to Table 3, our methods outperform existing methods to retain knowledge of concepts learned from previous tasks. In general the concepts are more stable in random class distribution since different tasks might share more overlapped concepts.

Table 9: The ratio of units which still detect the same concepts they detected in the last task for CIFAR-10 and CIFAR-100. Class labels are distributed randomly, and the results are average over three runs. Our methods outperform the existing works for preserving learned concepts.

| Method | CIFAR10, 5T | | | | CIFAR100, 5T | | | |
|---|---|---|---|---|---|---|---|---|
| | Task 2 | Task 3 | Task 4 | Task 5 | Task 2 | Task 3 | Task 4 | Task 5 |
| **Baseline** | | | | | | | | |
| Finetune | 0 | 0 | 0 | 0 | 0.020 | 0.005 | 0.010 | 0 |
| EWC | 0 | 0 | 0 | 0 | 0.065 | 0.010 | 0 | 0 |
| SI | 0 | 0 | 0 | 0 | 0 | 0 | 0 | 0 |
| LwF | 0 | 0 | 0 | 0 | 0.020 | 0.030 | 0 | 0 |
| GEM | 0 | 0.031 | 0 | 0 | 0.138 | 0.043 | 0.006 | 0.023 |
| MIR | 0 | 0 | 0 | 0 | 0.028 | 0 | 0.050 | 0.052 |
| **Ours** | | | | | | | | |
| CC-freeze-all | **0.052** | **0.500** | **0.111** | **0.200** | **0.578** | **0.636** | **0.782** | 0.741 |
| CC-freeze-part | 0.023 | 0.450 | 0.090 | 0.150 | 0.210 | 0.466 | 0.687 | **0.869** |

Table 10: The ratio of units which still detect the same concepts they detected in the last task for TinyImagenet. Class labels are distributed randomly, and the results are average over three runs. Our methods outperform the existing works for preserving learned concepts.

| Method | TinyImagenet, 5T | | | |
|---|---|---|---|---|
| | Task 2 | Task 3 | Task 4 | Task 5 |
| **Baseline** | | | | |
| Finetune | 0 | 0.010 | 0 | 0.030 |
| EWC | 0.030 | 0.015 | 0 | 0.010 |
| SI | 0.011 | 0 | 0 | 0 |
| LwF | 0.015 | 0 | 0 | 0 |
| GEM | 0 | 0.041 | 0.012 | 0.029 |
| MIR | 0 | 0 | 0 | 0 |
| **Ours** | | | | |
| CC-freeze-all | **0.724** | **0.833** | **0.741** | 0.812 |
| CC-freeze-part | 0.586 | 0.739 | 0.666 | **0.821** |

## A.8 DISCUSSION OF CONCEPT EVOLUTION FOR CC-CBM

In this section, we study the evolution of the concepts represented by neurons and final layer weights as we train CC-CBM across different tasks under 5-tasks scenario. We analyse a classes' final layer weights after learning its task, and after learning a new task. We visualize the final layer weights of Finetune-CBM and CC-CBM by Sankey diagrams, only including weights with absolute value greater than 0.05. Negative weights are reported as "NOT {concept}". The visualizations for random classes in three datasets are in Figure 5 6 7. Compared with Finetune-CBM, we can see CC-CBM is much better at retaining and using knowledge of concepts learned from previous tasks. This helps explain why CC-CBM performs better in $F_T$ and $\bar{F}_T$.

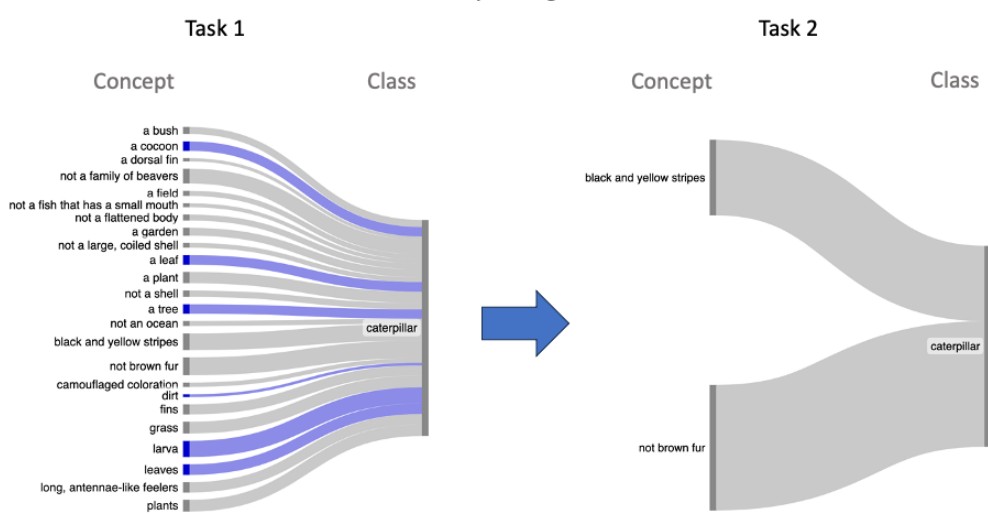

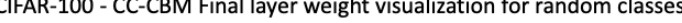

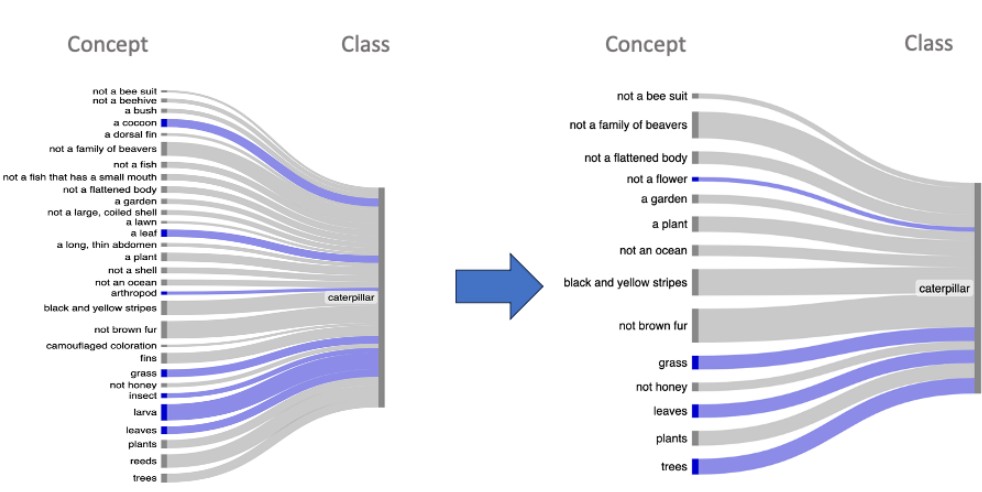

Figure 5: Final weight visualization for random classes in Finetune-CBM (top) and CC-CBM (bottom) trained on CIFAR-100 under 5-tasks scenario. We show the class "caterpillar"'s weight after training on task 1 and task 2. Concepts generated from the caterpillar class itself are colored blue, and other concepts from the original task for caterpillar are colored gray. The class distribution is in Table 26. We can see CC-CBM keeps a similar final layer while Finetune-CBM loses most significant weights.

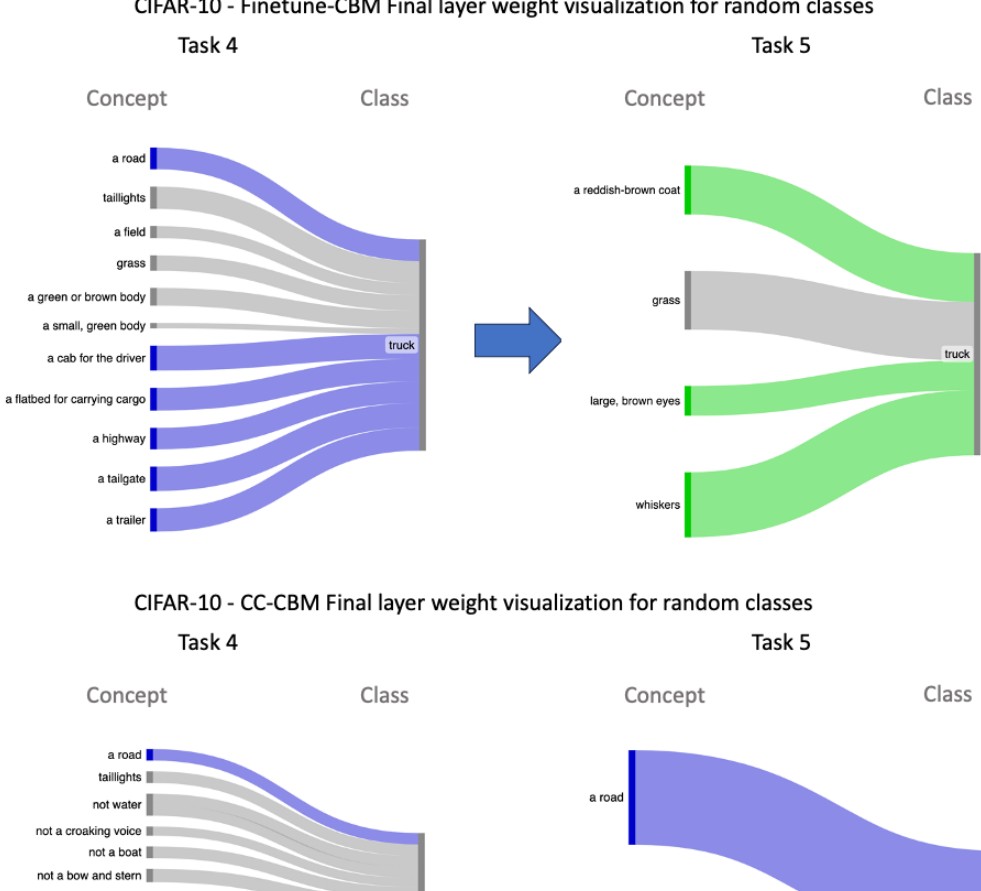

Figure 6: Final weight visualization for random classes in Finetune-CBM and CC-CBM trained on CIFAR-10 under 5-tasks scenario. We show the class "truck"'s weight after training on task 4 and task 5. Concepts generated from the truck class itself are colored blue, and other concepts from the original task for truck are colored gray. Concepts from the new task are colored green. The class distribution is in Table 23. We can see both models change significantly, but CC-CBM keeps more concepts from original task.

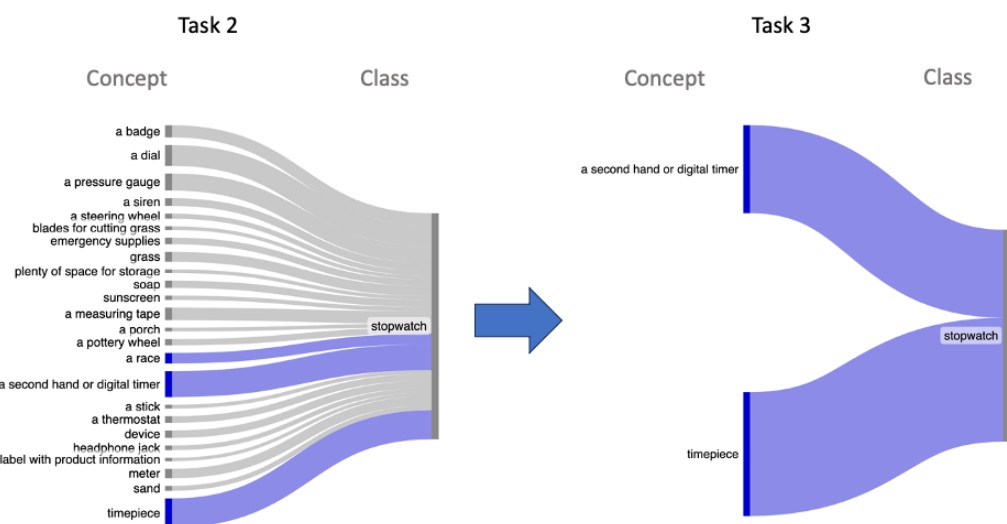

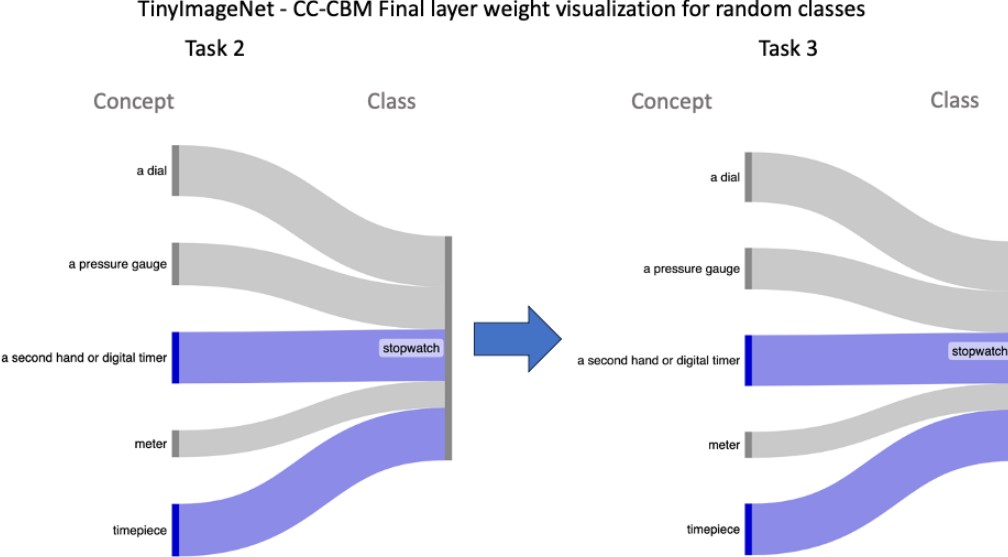

Figure 7: Final weight visualization for a random class in Finetune-CBM and CC-CBM trained on TinyImageNet under 5-tasks scenario. We show the class "stopwatch"'s weight after training on task 2 and task 3. Concepts generated from the stopwatch class itself are colored blue, and other concepts from the original task for stopwatch are colored gray. The class distribution is in Table 29. We can see CC-CBM weights stay almost identical, while Finetune-CBM changes a lot.

## A.9 5 TASK STANDARD METRIC RESULTS

Table 11 and 12 report comparisons for CC and CC-CBM under 5-tasks scenario respectively. Different from Table 1 and 2 that report $\bar{A}_T$ and $\bar{F}_T$, Table 11 and 12 report $A_T$ and $F_T$ that only reflect models' performance at the last stage.

Table 11: Accuracy comparison for CC. ↑ means larger values are better, while ↓ means smaller values are better. The **Improvement** is compared with the strongest baseline for each block.

| | CIFAR-10, 5T | | CIFAR-100, 5T | | TinyImagenet, 5T | |
|---|---|---|---|---|---|---|
| | $A_T \uparrow$ | $F_T \downarrow$ | $A_T \uparrow$ | $F_T \downarrow$ | $A_T \uparrow$ | $F_T \downarrow$ |
| **Baseline in Category (i)** | | | | | | |
| Finetune | $15.89 \pm 0.76$ | $78.54 \pm 0.87$ | $16.48 \pm 2.63$ | $56.65 \pm 3.24$ | $13.20 \pm 2.23$ | $43.50 \pm 0.45$ |
| EWC | $18.72 \pm 0.79$ | $75.66 \pm 2.26$ | $15.79 \pm 1.58$ | $55.79 \pm 3.87$ | $13.07 \pm 3.55$ | $43.04 \pm 2.41$ |
| SI | $18.96 \pm 0.35$ | $74.19 \pm 1.66$ | $13.30 \pm 2.62$ | $\mathbf{50.59 \pm 3.50}$ | $10.51 \pm 3.01$ | $39.19 \pm 3.91$ |
| LwF | $19.01 \pm 0.23$ | $75.00 \pm 1.33$ | $\mathbf{17.92 \pm 0.10}$ | $56.85 \pm 2.54$ | $13.47 \pm 1.48$ | $43.75 \pm 1.15$ |
| **Baseline in Category (ii)** | | | | | | |
| Adam-NSCL | $20.60 \pm 0.71$ | $76.82 \pm 1.72$ | $17.70 \pm 0.85$ | $72.35 \pm 1.25$ | $\mathbf{14.15 \pm 2.19}$ | $62.71 \pm 2.93$ |
| **Ours** | | | | | | |
| CC-freeze-all | $\mathbf{21.14 \pm 1.90}$ | $\mathbf{70.68 \pm 0.81}$ | $14.66 \pm 1.81$ | $51.17 \pm 2.48$ | $11.20 \pm 0.48$ | $\mathbf{37.85 \pm 0.83}$ |
| CC-freeze-part | $20.71 \pm 0.84$ | $71.42 \pm 1.12$ | $15.17 \pm 3.47$ | $52.99 \pm 2.85$ | $11.34 \pm 3.67$ | $38.53 \pm 0.69$ |
| **Improvement** | **0.54** | **3.51** | -2.75 | -0.58 | -2.81 | **1.34** |
| **Baseline in Category (iii)** | | | | | | |
| GEM | $22.51 \pm 0.05$ | $73.50 \pm 3.80$ | $19.07 \pm 0.62$ | $52.87 \pm 2.74$ | $11.12 \pm 1.79$ | $37.58 \pm 1.15$ |
| MIR | $28.18 \pm 2.34$ | $47.71 \pm 0.99$ | $9.75 \pm 3.39$ | $43.09 \pm 0.68$ | $10.91 \pm 3.89$ | $38.42 \pm 1.57$ |
| DER | $28.85 \pm 2.34$ | $70.81 \pm 0.99$ | $19.74 \pm 2.28$ | $54.55 \pm 1.60$ | $11.35 \pm 1.68$ | $47.39 \pm 2.26$ |
| **Ours** | | | | | | |
| CC-freeze-all-GEM | $24.66 \pm 0.62$ | $68.96 \pm 0.66$ | $\mathbf{25.68 \pm 1.72}$ | $\mathbf{36.83 \pm 0.26}$ | $9.89 \pm 2.56$ | $\mathbf{33.45 \pm 3.05}$ |
| CC-freeze-part-GEM | $23.22 \pm 0.96$ | $72.18 \pm 2.00$ | $23.70 \pm 3.84$ | $43.73 \pm 3.81$ | $10.91 \pm 2.26$ | $36.37 \pm 2.35$ |
| CC-freeze-all-MIR | $33.16 \pm 1.54$ | $\mathbf{42.00 \pm 2.57}$ | $14.49 \pm 0.05$ | $43.82 \pm 0.12$ | $12.29 \pm 0.60$ | $38.42 \pm 1.78$ |
| CC-freeze-part-MIR | $\mathbf{34.88 \pm 1.37}$ | $42.11 \pm 2.04$ | $14.57 \pm 2.99$ | $44.41 \pm 1.38$ | $\mathbf{12.47 \pm 1.02}$ | $38.50 \pm 3.76$ |
| **Improvement** | **6.03** | **5.71** | **5.94** | **6.26** | **1.12** | **4.13** |

Table 12: Accuracy comparison for CC-CBM. All models are pre-trained on the Place365 dataset Zhou et al. (2017). ↑ means larger values are better, while ↓ means smaller values are better. The **Improvement** is compared with the strongest baseline for each block.

| | CIFAR-10, 5T | | CIFAR-100, 5T | | TinyImagenet, 5T | |
|---|---|---|---|---|---|---|
| | $A_T \uparrow$ | $F_T \downarrow$ | $A_T \uparrow$ | $F_T \downarrow$ | $A_T \uparrow$ | $F_T \downarrow$ |
| **Baseline in Category (i)** | | | | | | |
| Finetune | $17.99 \pm 0.28$ | $79.66 \pm 0.78$ | $17.65 \pm 2.55$ | $64.94 \pm 2.15$ | $15.34 \pm 3.83$ | $54.81 \pm 0.71$ |
| EWC | $19.00 \pm 0.21$ | $76.92 \pm 0.78$ | $17.44 \pm 3.98$ | $64.43 \pm 1.84$ | $15.07 \pm 2.23$ | $54.77 \pm 0.54$ |
| SI | $19.31 \pm 0.72$ | $74.78 \pm 1.70$ | $17.12 \pm 3.89$ | $63.11 \pm 1.03$ | $14.47 \pm 1.02$ | $53.69 \pm 1.32$ |
| LwF | $19.11 \pm 0.31$ | $76.56 \pm 0.84$ | $\mathbf{19.98 \pm 0.64}$ | $63.79 \pm 2.65$ | $\mathbf{17.08 \pm 1.50}$ | $55.40 \pm 1.85$ |
| **Baseline in Category (ii)** | | | | | | |
| Adam-NSCL | $21.60 \pm 0.48$ | $77.73 \pm 0.86$ | $16.23 \pm 3.39$ | $66.26 \pm 3.18$ | $13.44 \pm 1.70$ | $58.86 \pm 0.70$ |
| **Ours** | | | | | | |
| Finetune-CBM | $21.14 \pm 0.67$ | $69.47 \pm 2.22$ | $16.43 \pm 2.82$ | $55.76 \pm 3.55$ | $14.91 \pm 0.65$ | $51.39 \pm 3.78$ |
| CC-CBM | $\mathbf{22.16 \pm 0.76}$ | $\mathbf{67.92 \pm 0.30}$ | $16.37 \pm 3.82$ | $\mathbf{45.71 \pm 2.55}$ | $15.68 \pm 1.84$ | $\mathbf{39.40 \pm 2.86}$ |
| **Improvement** | **0.56** | **6.86** | -3.55 | **17.40** | -1.40 | **14.29** |
| **Baseline in Category (iii)** | | | | | | |
| GEM | $24.11 \pm 1.86$ | $59.84 \pm 2.88$ | $20.67 \pm 1.75$ | $63.49 \pm 3.15$ | $5.64 \pm 0.38$ | $37.52 \pm 0.02$ |
| MIR | $\mathbf{33.99 \pm 2.92}$ | $48.50 \pm 1.94$ | $8.33 \pm 0.64$ | $\mathbf{46.33 \pm 2.06}$ | $2.97 \pm 3.69$ | $32.09 \pm 0.53$ |
| DER | $28.88 \pm 2.14$ | $75.42 \pm 0.99$ | $17.89 \pm 0.43$ | $66.93 \pm 1.39$ | $\mathbf{13.64 \pm 2.09}$ | $57.98 \pm 2.48$ |
| **Ours** | | | | | | |
| CC-CBM-GEM | $28.93 \pm 1.74$ | $61.86 \pm 2.71$ | $\mathbf{23.64 \pm 3.82}$ | $58.54 \pm 3.75$ | $5.37 \pm 1.67$ | $\mathbf{29.06 \pm 3.74}$ |
| CC-CBM-MIR | $33.94 \pm 1.77$ | $\mathbf{38.93 \pm 1.89}$ | $13.81 \pm 1.94$ | $54.76 \pm 1.59$ | $5.07 \pm 3.66$ | $34.75 \pm 1.54$ |
| **Improvement** | -0.05 | **9.57** | **2.97** | -8.43 | -8.27 | **3.03** |

## A.10 10, 20 TASKS EXPERIMENT RESULTS

Table 13 reports comparison for CC under 10-tasks scenario for CIFAR-100 and TinyImagenet. Compared with exemplar-free baselines, CC generally forgets less as $F_T$ and $\bar{F}_T$ shown. However, CC sometimes is worse in $A_T$ and $\bar{A}_T$, which is the limitation of our methods. We view this as the future works to be improved, since this paper's goal is to retain intepretable concepts in models. Nevertheless, CC generally improves exemplar-based baselines when combined with them, which is same as 5-tasks scenario.

Table 14 reports comparison for CC-CBM under 10-tasks scenario for CIFAR-100 and TinyImagenet. Similar to CC, CC-CBM performs well in forgetting metrics $F_T$ and $\bar{F}_T$, but performs worse in $A_T$ and $\bar{A}_T$ when comparing with exemplar-free baselines. We believes this is partially because LF-CBM's training procedure is different than standard end-to-end training. As LF-CBM Oikarinen et al. (2023) paper shown, LF-CBM's classification accuracy is usually worse than standard model's. Again, we view this as future works to be improved. Besides, CC-CBM still can improve exemplar-based baselines when combine with them.

Table 15 and 16 report comparison for CC and CC-CBM under 20-tasks scenario in CIFAR-100 respectively. Similar to 10-tasks scenario, our methods can improve exemplar-based baselines when combine with them, while the comparison with exemplar-free methods is still under the same trend.

Table 13: Accuracy comparison for CC. ↑ means larger values are better, while ↓ means smaller values are better. The **Improvement** is compared with the strongest baseline for each block.

| | CIFAR-100, 10T | | | | TinyImagenet, 10T | | | |
|---|---|---|---|---|---|---|---|---|
| | $A_T \uparrow$ | $F_T \downarrow$ | $\bar{A}_T \uparrow$ | $\bar{F}_T \downarrow$ | $A_T \uparrow$ | $F_T \downarrow$ | $\bar{A}_T \uparrow$ | $\bar{F}_T \downarrow$ |
| **Baseline in Category (i)** | | | | | | | | |
| Finetune | $6.49 \pm 1.23$ | $65.31 \pm 1.32$ | $13.87 \pm 0.83$ | $64.20 \pm 1.52$ | $5.15 \pm 2.13$ | $49.07 \pm 0.57$ | $10.23 \pm 1.35$ | $45.83 \pm 0.21$ |
| EWC | $6.71 \pm 0.63$ | $65.32 \pm 1.12$ | $13.88 \pm 0.45$ | $63.90 \pm 0.94$ | $5.48 \pm 2.35$ | $47.42 \pm 1.47$ | $10.11 \pm 2.12$ | $45.19 \pm 1.29$ |
| SI | $6.74 \pm 1.17$ | $66.07 \pm 1.91$ | $14.01 \pm 1.37$ | $64.39 \pm 1.33$ | $5.39 \pm 1.57$ | $\mathbf{44.68 \pm 1.40}$ | $10.33 \pm 1.32$ | $46.21 \pm 1.36$ |
| LwF | $7.85 \pm 2.21$ | $64.13 \pm 2.19$ | $\mathbf{15.02 \pm 2.85}$ | $62.66 \pm 1.25$ | $5.95 \pm 2.35$ | $52.17 \pm 1.26$ | $10.82 \pm 2.35$ | $45.57 \pm 1.36$ |
| **Baseline in Category (ii)** | | | | | | | | |
| Adam-NSCL | $\mathbf{7.95 \pm 2.77}$ | $62.54 \pm 1.89$ | $13.69 \pm 2.15$ | $\mathbf{61.54 \pm 2.40}$ | $\mathbf{6.63 \pm 2.23}$ | $49.43 \pm 1.45$ | $10.63 \pm 1.33$ | $49.43 \pm 1.45$ |
| **Ours** | | | | | | | | |
| CC-freeze-all | $6.99 \pm 1.20$ | $\mathbf{62.23 \pm 0.36}$ | $13.81 \pm 1.45$ | $62.79 \pm 1.70$ | $5.76 \pm 2.67$ | $48.89 \pm 2.17$ | $11.24 \pm 0.34$ | $49.39 \pm 1.92$ |
| CC-freeze-part | $6.97 \pm 1.26$ | $64.03 \pm 1.32$ | $13.71 \pm 1.36$ | $63.28 \pm 1.04$ | $5.86 \pm 0.56$ | $49.99 \pm 2.16$ | $\mathbf{11.25 \pm 1.28}$ | $\mathbf{45.00 \pm 1.20}$ |
| **Improvement** | -0.96 | 0.31 | -1.21 | -1.74 | -0.77 | -4.21 | 0.43 | 0.19 |
| **Baseline in Category (iii)** | | | | | | | | |
| GEM | $8.25 \pm 1.07$ | $60.31 \pm 4.25$ | $15.71 \pm 1.35$ | $58.77 \pm 2.42$ | $6.00 \pm 2.34$ | $47.13 \pm 1.25$ | $10.74 \pm 2.62$ | $47.68 \pm 1.67$ |
| MIR | $4.64 \pm 0.37$ | $59.37 \pm 1.96$ | $12.61 \pm 1.02$ | $60.51 \pm 1.72$ | $4.54 \pm 0.82$ | $46.19 \pm 2.43$ | $10.11 \pm 1.34$ | $46.19 \pm 2.17$ |
| DER | $7.67 \pm 0.93$ | $64.86 \pm 2.84$ | $13.81 \pm 2.03$ | $63.76 \pm 2.19$ | $\mathbf{6.49 \pm 0.82}$ | $57.60 \pm 2.43$ | $\mathbf{13.10 \pm 1.34}$ | $55.95 \pm 3.64$ |
| **Ours** | | | | | | | | |
| CC-freeze-all-GEM | $\mathbf{15.61 \pm 1.02}$ | $\mathbf{52.01 \pm 1.34}$ | $\mathbf{29.83 \pm 2.15}$ | $\mathbf{52.43 \pm 2.36}$ | $3.03 \pm 3.05$ | $\mathbf{22.39 \pm 3.91}$ | $7.75 \pm 2.90$ | $34.75 \pm 2.37$ |
| CC-freeze-part-GEM | $7.65 \pm 1.85$ | $60.12 \pm 1.37$ | $14.84 \pm 1.39$ | $54.12 \pm 2.73$ | $2.97 \pm 2.34$ | $24.91 \pm 2.02$ | $5.52 \pm 2.40$ | $\mathbf{30.44 \pm 2.59}$ |
| CC-freeze-all-MIR | $5.43 \pm 1.54$ | $52.22 \pm 1.65$ | $12.86 \pm 1.08$ | $56.19 \pm 2.30$ | $5.09 \pm 3.21$ | $41.34 \pm 1.28$ | $9.97 \pm 2.41$ | $43.21 \pm 1.32$ |
| CC-freeze-part-MIR | $5.42 \pm 1.34$ | $52.34 \pm 1.37$ | $12.89 \pm 0.62$ | $56.72 \pm 1.84$ | $5.17 \pm 2.58$ | $42.87 \pm 0.99$ | $9.99 \pm 1.23$ | $43.23 \pm 1.41$ |
| **Improvement** | 7.36 | 7.36 | 14.12 | 6.34 | -1.32 | 24.74 | -3.11 | 15.75 |

Table 14: Accuracy comparison for CC-CBM. All models are pre-trained on the Place365 dataset Zhou et al. (2017). ↑ means larger values are better, while ↓ means smaller values are better. The **Improvement** is compared with the strongest baseline for each block.

| | CIFAR-100, 10T | | | | TinyImagenet, 10T | | | |
|---|---|---|---|---|---|---|---|---|
| | $A_T \uparrow$ | $F_T \downarrow$ | $\bar{A}_T \uparrow$ | $\bar{F}_T \downarrow$ | $A_T \uparrow$ | $F_T \downarrow$ | $\bar{A}_T \uparrow$ | $\bar{F}_T \downarrow$ |
| **Baseline in Category (i)** | | | | | | | | |
| Finetune | $8.09 \pm 1.82$ | $80.43 \pm 1.42$ | $17.34 \pm 1.87$ | $81.23 \pm 2.30$ | $4.61 \pm 0.18$ | $66.40 \pm 3.42$ | $14.62 \pm 0.58$ | $70.70 \pm 1.56$ |
| EWC | $7.59 \pm 1.32$ | $80.99 \pm 1.28$ | $17.23 \pm 0.96$ | $81.08 \pm 2.34$ | $5.62 \pm 2.38$ | $67.27 \pm 1.30$ | $14.88 \pm 1.28$ | $71.25 \pm 1.38$ |
| SI | $8.54 \pm 1.29$ | $81.43 \pm 1.83$ | $17.53 \pm 0.65$ | $81.60 \pm 2.45$ | $7.90 \pm 1.40$ | $71.59 \pm 0.82$ | $15.65 \pm 1.19$ | $72.33 \pm 2.08$ |
| LwF | $13.83 \pm 1.26$ | $77.21 \pm 1.53$ | $24.99 \pm 2.29$ | $71.19 \pm 3.33$ | $10.40 \pm 2.35$ | $71.40 \pm 1.59$ | $20.42 \pm 1.86$ | $66.78 \pm 3.95$ |
| **Baseline in Category (ii)** | | | | | | | | |
| Adam-NSCL | $14.29 \pm 1.19$ | $69.75 \pm 3.15$ | $23.19 \pm 1.38$ | $76.24 \pm 1.29$ | $9.74 \pm 2.23$ | $66.87 \pm 1.45$ | $18.63 \pm 2.45$ | $69.97 \pm 1.24$ |
| **Ours** | | | | | | | | |
| Finetune-CBM | $7.31 \pm 0.82$ | $68.88 \pm 1.40$ | $14.91 \pm 1.28$ | $66.72 \pm 2.73$ | $7.18 \pm 1.26$ | $65.76 \pm 1.42$ | $14.41 \pm 1.89$ | $64.64 \pm 3.02$ |
| CC-CBM | $7.34 \pm 1.68$ | $37.90 \pm 1.59$ | $14.98 \pm 1.09$ | $61.96 \pm 1.73$ | $7.37 \pm 2.54$ | $56.80 \pm 1.27$ | $14.98 \pm 2.93$ | $60.02 \pm 3.09$ |
| **Improvement** | -6.95 | **31.85** | -10.01 | **9.23** | -3.03 | **9.60** | -5.44 | **6.76** |
| **Baseline in Category (iii)** | | | | | | | | |
| GEM | $11.47 \pm 1.32$ | $66.44 \pm 3.14$ | $22.21 \pm 1.20$ | $70.54 \pm 2.32$ | $4.12 \pm 1.79$ | $54.29 \pm 1.62$ | $13.44 \pm 1.33$ | $63.58 \pm 1.11$ |
| MIR | $3.24 \pm 2.09$ | $58.77 \pm 1.69$ | $14.19 \pm 2.21$ | $73.32 \pm 0.93$ | $3.37 \pm 0.84$ | $58.83 \pm 2.06$ | $9.40 \pm 3.29$ | $59.29 \pm 4.20$ |
| DER | $7.52 \pm 1.95$ | $78.51 \pm 1.47$ | $17.04 \pm 1.29$ | $80.78 \pm 1.92$ | $8.29 \pm 0.59$ | $72.69 \pm 1.98$ | $16.64 \pm 0.93$ | $72.00 \pm 2.48$ |
| **Ours** | | | | | | | | |
| CC-CBM-GEM | $8.68 \pm 2.41$ | $53.13 \pm 3.71$ | $22.46 \pm 1.15$ | $59.36 \pm 2.33$ | $6.20 \pm 3.07$ | $47.95 \pm 1.48$ | $14.96 \pm 3.48$ | $57.94 \pm 1.26$ |
| CC-CBM-MIR | $8.38 \pm 1.35$ | $71.02 \pm 2.06$ | $16.65 \pm 2.23$ | $70.49 \pm 1.90$ | $7.13 \pm 2.15$ | $60.98 \pm 2.38$ | $14.07 \pm 2.31$ | $61.14 \pm 2.36$ |
| CC-CBM-DER | $9.63 \pm 2.17$ | $78.52 \pm 1.98$ | $23.29 \pm 2.92$ | $69.05 \pm 2.17$ | $8.37 \pm 2.73$ | $70.85 \pm 4.05$ | $16.90 \pm 3.01$ | $71.56 \pm 2.38$ |
| **Improvement** | -1.84 | **5.64** | **1.08** | **11.18** | **0.08** | **6.34** | **0.26** | **1.35** |

Table 15: Accuracy comparison for CC. ↑ means larger values are better, while ↓ means smaller values are better. The **Improvement** is compared with the strongest baseline for each block.

| | CIFAR-100, 20T | | | |
|---|---|---|---|---|
| | $A_T \uparrow$ | $F_T \downarrow$ | $\bar{A}_T \uparrow$ | $\bar{F}_T \downarrow$ |
| **Baseline in Category (i)** | | | | |
| Finetune | $3.80 \pm 2.26$ | $69.05 \pm 2.23$ | $9.72 \pm 3.54$ | $70.25 \pm 0.88$ |
| EWC | $3.56 \pm 0.16$ | $68.67 \pm 0.71$ | $9.72 \pm 2.72$ | $70.11 \pm 3.29$ |
| SI | $3.92 \pm 2.37$ | $76.58 \pm 3.77$ | $10.61 \pm 1.65$ | $76.19 \pm 0.00$ |
| LwF | $4.14 \pm 2.12$ | $77.06 \pm 1.48$ | $10.54 \pm 3.72$ | $75.25 \pm 2.74$ |
| **Baseline in Category (ii)** | | | | |
| Adam-NSCL | $10.59 \pm 0.99$ | $67.32 \pm 3.52$ | $19.45 \pm 2.85$ | $68.21 \pm 2.80$ |
| **Ours** | | | | |
| CC-freeze-all | $3.35 \pm 1.13$ | $63.12 \pm 3.02$ | $8.82 \pm 2.43$ | $65.21 \pm 2.34$ |
| CC-freeze-part | $3.79 \pm 2.52$ | $72.40 \pm 3.61$ | $9.99 \pm 2.24$ | $72.52 \pm 1.42$ |
| **Improvement** | -6.80 | **4.20** | -9.46 | **3.00** |
| **Baseline in Category (iii)** | | | | |
| GEM | $14.90 \pm 3.79$ | $70.56 \pm 3.06$ | $24.88 \pm 3.34$ | $66.55 \pm 0.44$ |
| MIR | $11.32 \pm 2.07$ | $45.31 \pm 0.08$ | $17.49 \pm 3.68$ | $57.71 \pm 2.67$ |
| DER | $12.59 \pm 1.41$ | $56.85 \pm 0.42$ | $18.60 \pm 1.19$ | $65.39 \pm 0.59$ |
| **Ours** | | | | |
| CC-freeze-all-GEM | $20.72 \pm 0.88$ | $35.48 \pm 2.14$ | $27.73 \pm 2.84$ | $66.51 \pm 1.72$ |
| CC-freeze-part-GEM | $17.40 \pm 3.74$ | $63.10 \pm 2.51$ | $24.18 \pm 2.34$ | $68.03 \pm 2.30$ |
| CC-freeze-all-MIR | $12.75 \pm 0.68$ | $39.36 \pm 2.46$ | $17.17 \pm 2.60$ | $41.74 \pm 0.31$ |
| CC-freeze-part-MIR | $13.03 \pm 1.71$ | $52.47 \pm 0.30$ | $18.02 \pm 3.30$ | $53.93 \pm 3.61$ |
| **Improvement** | **5.82** | **9.83** | **2.85** | **15.97** |

Table 16: Accuracy comparison for CC-CBM. All models are pre-trained on the Place365 dataset Zhou et al. (2017). ↑ means larger values are better, while ↓ means smaller values are better. The **Improvement** is compared with the strongest baseline for each block.

| | CIFAR-100, 20T | | | |
|---|---|---|---|---|
| | $A_T \uparrow$ | $F_T \downarrow$ | $\bar{A}_T \uparrow$ | $\bar{F}_T \downarrow$ |
| **Baseline in Category (i)** | | | | |
| Finetune | $4.03 \pm 0.60$ | $71.17 \pm 0.86$ | $10.35 \pm 1.38$ | $77.07 \pm 3.09$ |
| EWC | $3.92 \pm 3.23$ | $73.90 \pm 2.41$ | $10.65 \pm 1.20$ | $78.75 \pm 3.67$ |
| SI | $3.73 \pm 0.45$ | $78.96 \pm 0.36$ | $11.21 \pm 2.85$ | $82.57 \pm 3.67$ |
| LwF | $5.23 \pm 3.18$ | $85.31 \pm 2.62$ | $13.32 \pm 3.94$ | $85.45 \pm 3.49$ |
| **Baseline in Category (ii)** | | | | |
| Adam-NSCL | $\mathbf{10.93 \pm 1.33}$ | $69.52 \pm 0.54$ | $\mathbf{17.96 \pm 0.74}$ | $87.83 \pm 2.50$ |
| **Ours** | | | | |
| Finetune-CBM | $4.05 \pm 3.87$ | $77.40 \pm 3.48$ | $10.67 \pm 0.05$ | $74.86 \pm 3.00$ |
| CC-CBM | $4.04 \pm 2.87$ | $\mathbf{66.55 \pm 3.23}$ | $11.48 \pm 1.18$ | $\mathbf{73.06 \pm 0.29}$ |
| **Improvement** | -6.88 | **2.97** | -6.48 | **4.01** |
| **Baseline in Category (iii)** | | | | |
| GEM | $12.59 \pm 2.84$ | $58.33 \pm 2.69$ | $22.94 \pm 0.34$ | $67.12 \pm 2.08$ |
| MIR | $11.01 \pm 1.01$ | $\mathbf{29.39 \pm 2.81}$ | $16.68 \pm 0.41$ | $\mathbf{51.33 \pm 1.35}$ |
| DER | $11.65 \pm 3.07$ | $55.93 \pm 3.49$ | $18.86 \pm 2.72$ | $69.70 \pm 3.52$ |
| **Ours** | | | | |
| CC-CBM-GEM | $\mathbf{12.87 \pm 2.37}$ | $61.74 \pm 3.69$ | $\mathbf{25.62 \pm 2.83}$ | $68.64 \pm 0.53$ |
| CC-CBM-MIR | $12.53 \pm 0.08$ | $36.31 \pm 1.58$ | $19.55 \pm 1.90$ | $55.79 \pm 1.79$ |
| **Improvement** | **0.28** | -6.92 | **2.68** | -4.46 |

### A.11 COMPUTATIONAL EFFICIENCY

Table 17 shows the maximum GPU usage of Finetune, EWC, GEM and our CC on CIFAR-10 under 5-tasks scenario. Even integrated with the interpretable tool CLIP-Dissect (Oikarinen & Weng, 2022), CC's maximum GPU usage is still lower than EWC and GEM. The experiment results show our methods are efficient.

Table 17: Maximum GPU usage comparison for our CC, under CIFAR-10 5-tasks scenario. CC's maximum GPU usage is smaller than EWC and GEM, which shows CC's computational efficiency.

| Method | Finetune | EWC | GEM | CC-freeze-all/ CC-freeze-part |
|---|---|---|---|---|
| Max GPU usage (GB) | 5.51 | 5.65 | 5.68 | 5.61 |

### A.12 COMPARISON WITH SSRE

We compare our methods with SSRE (Zhu et al., 2022). Since we can not access the pre-training details described in SSRE paper, we compare our CC with SSRE which is trained from scratch. Besides metrics described in Section 5, we also measure **Learning Accuracy** ($F_T = \frac{1}{T} \sum_{i=1}^{T} a_{i,i}$) (Mirzadeh et al., 2022b; Riemer et al., 2018; Yin et al., 2021; Mirzadeh et al., 2022a) to understand model's ability to learn new tasks.

Experiment results are in Table 18 and 19. CC forgets more as the performace in $F_T$ is worse. However, it has much better learning ability as $L_T$ shown.

Table 18: Comparison between CC and SSRE Zhu et al. (2022) in CIFAR-10 and CIFAR-100. ↑ means larger values are better, while ↓ means smaller values are better. Even though CC forgets more, it has better learning ability.

| | CIFAR-10, 5T | | | CIFAR-100, 5T | | | CIFAR-100, 10T | | | CIFAR-100, 20T | | |
|---|---|---|---|---|---|---|---|---|---|---|---|---|
| | $A_T \uparrow$ | $F_T \downarrow$ | $L_T \uparrow$ | $A_T \uparrow$ | $F_T \downarrow$ | $L_T \uparrow$ | $A_T \uparrow$ | $F_T \downarrow$ | $L_T \uparrow$ | $A_T \uparrow$ | $F_T \downarrow$ | $L_T \uparrow$ |
| SSRE | **28.44** | **15.96** | 36.18 | **31.55** | **7.97** | 32.73 | **17.72** | **7.46** | 11.29 | **8.94** | **6.73** | 4.51 |
| CC-freezed-all | 21.14 | 70.68 | **93.29** | 16.66 | 51.17 | **63.22** | 6.99 | 62.23 | **64.25** | 3.35 | 63.12 | **66.46** |

Table 19: Comparison between CC and SSRE Zhu et al. (2022) in TinyImagenet. ↑ means larger values are better, while ↓ means smaller values are better. Even though CC forgets more, it has better learning ability.

| | TinyImagenet, 5T | | | TinyImagenet, 10T | | |
|---|---|---|---|---|---|---|
| | $A_T \uparrow$ | $F_T \downarrow$ | $L_T \uparrow$ | $A_T \uparrow$ | $F_T \downarrow$ | $L_T \uparrow$ |
| SSRE | **13.76** | **6.06** | 13.59 | **11.44** | **7.33** | 15.83 |
| CC-freezed-all | 13.20 | 37.85 | **47.92** | 5.76 | 48.89 | **49.81** |

## A.13 COMPARISON WITH ICICLE

We compare CC-CBM with ICICLE (Rymarczyk et al., 2023) in the CUB200 (Wah et al., 2011) dataset. CUB200 is a bird species classification dataset that has 200 classes with 5994 training examples and 5794 testing examples. The experiment results are in Table 20. Our method outperforms ICICLE by 2.02% in $\bar{A}_T$ and 31.68% in $\bar{F}_T$.

Table 20: Comparison between CC-CBM and ICICLE (Rymarczyk et al., 2023) and other baselines in CUB200 5-task scenario . ↑ means larger values are better, while ↓ means smaller values are better. CC-CBM outperform ICICLE in both $\bar{A}_T$ and $\bar{F}_T$.

|  | CUB200, 5T | |
| --- | --- | --- |
|  | $\bar{A}_T \uparrow$ | $\bar{F}_T \downarrow$ |
| **Baseline in Category (i) and (ii)** | | |
| Finetune | 10.16 | 60.73 |
| EWC | 11.02 | 62.43 |
| LwF | 12.50 | 64.50 |
| Adam-NSCL | 13.27 | 40.73 |
| ICICLE | 11.40 | 53.66 |
| **Ours** | | |
| CC-CBM | **13.42** | **21.98** |

## A.14 DETAILS OF CONCEPT CONTROLLER

In this section, we study the technical details of the **Concept Controller**. ImageNet-10 is the 10-class subset of ImageNet (Russakovsky et al., 2015). For CC, Figure 8 shows the subnetwork sizes in different datasets. Generally, a more complicated dataset has a bigger subnetwork, but the size is adjustable by tuning hyperparameters.

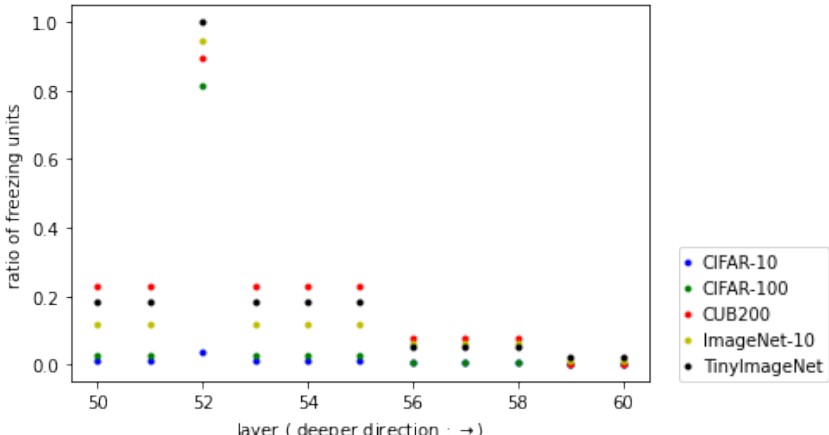

Figure 8: The subnetwork sizes in the last residual block after learning two tasks in the 5-task scenario. A more complicated dataset will have bigger subnetworks.

For CC-CBM, Table 21 and 22 show the number of concepts added per tasks. Overall, larger and more complicated datasets have more concepts.

We build CC-CBM based on LF-CBM Oikarinen et al. (2023). Therefore, we would like to discuss two properties related to LF-CBM. First, following LF-CBM's procedure, we use CLIP to calculate the activation matrix $P$ as described in Section 2.2. We can replace CLIP with other vision language aligned (VL-aligned) models as long as they have a text encoder and an image encoder to calculate matrix $P$. One future work will be replacing CLIP with other suitable VL-aligned models. Second, using GPT-3 to general concept sets is firstly proposed from an earlier work LF-CBM, which has many concurrent and follow-up works that use language models to create concept bottleneck layers (Yang et al., 2023b; Yan et al., 2023). Introducing additional information is widely exploited when designing CBM (Yuksekgonul et al., 2022; Zhou et al., 2018; Losch et al., 2019), and is reasonable task-relevant information that we can leverage. Meanwhile, using additional information to enhance a model's performance is common in other fields as well (Hu et al., 2021; Yang et al., 2023a). Even though LF-CBM has a potential leakage problem when using addtional text information, our method is still under the setting of continual learning. This is because during the training phase, the information for each class only appears in one task. One future work is to handle the potential leaking of CBM.

Table 21: Average concepts added per task in the 5-task scenario.

|  | CIFAR-10 | CIFAR-100 | CUB200 | ImageNet-10 | TinyImageNet |
|---|---|---|---|---|---|
| Number of Concept | 37.4 | 133.1 | 276.6 | 35.6 | 301.8 |

Table 22: Average concepts added per task in the 10-task scenario.

|  | CIFAR-100 | TinyImageNet |
|---|---|---|
| Number of Concept | 154.6 | 284.3 |

## A.15 CLASS DISTRIBUTION

Table 23: Classes distribution of CIFAR-10 separated by random seed 3456.

| Task 1 | automobile, dog |
|--------|-----------------|
| Task 2 | deer, horse |
| Task 3 | bird, frog |
| Task 4 | ship, truck |
| Task 5 | airplane, cat |

Table 24: Classes distribution of CIFAR-100 separated by random seed 3456.

| | |
|--------|------------------------------------------------------------------|
| Task 1 | bear,bee,butterfly,camel,caterpillar,chair,elephant, forest,hamster,lion, motorcycle,otter,plates, sea,shark,shrew,spider,tank,train,willow |
| Task 2 | beaver,bowls,boy,bridge,castle,cockroach,couch, dinosaur,house,keyboard, lawn mower,mushrooms,pears, pickup truck,poppies,possum,ray,skyscraper,wardrobe,whale |
| Task 3 | beetle,cloud,crocodile,lamp,leopard,lizard,palm,pine,porcupine,snail, streetcar,sweet peppers,table,telephone,television,tiger,tulips,turtle,woman,worm |
| Task 4 | baby,bed,bicycle,bottles,cans,chimpanzee,crab,lobster,man,maple, mouse,oak,orchids,plain,road,rocket,roses,skunk,squirrel,trout |
| Task 5 | apples,aquarium fish,bus,cattle,clock,cups,dolphin,flatfish,fox, girl, kangaroo,mountain,oranges,rabbit,raccoon,seal,snake,sunflowers,tractor,wolf |

Table 25: Classes distribution of CIFAR-100 separated by random seed 5678.

| | |
|--------|------------------------------------------------------------------|
| Task 1 | bowls, butterfly, cans, cloud, crab, girl, keyboard, leopard, lizard, mountain, mushrooms, pickup truck, roses, seal, snail, spider, squirrel, sunflowers, train, trout |
| Task 2 | baby, bear, bridge, cattle, chimpanzee, cockroach, couch, crocodile, flatfish, lamp, lobster, orchids, palm, plates, sea, shark, shrew, skyscraper, tulips, whale |
| Task 3 | apples, bed, bicycle, bus, caterpillar, cups, elephant, fox, hamster, kangaroo, lawn mower, maple, oak, plain, porcupine, rabbit, road, sweet peppers, tiger, wolf |
| Task 4 | bee, bottles, boy, chair, dinosaur, house, oranges, otter, pine, poppies, ray, snake, tank, telephone, tractor, turtle, wardrobe, willow, woman, worm |
| Task 5 | aquarium fish, beaver, beetle, camel, castle, clock, dolphin, forest, lion, man, motorcycle, mouse, pears, possum, raccoon, rocket, skunk, streetcar, table, television |

Table 26: Classes distribution of CIFAR-100 separated by grouping similar classes together.

| Task 1 | beaver , dolphin , otter , seal , whale , aquarium fish , flatfish , ray , shark , trout , bee , beetle , butterfly , caterpillar , cockroach , crab , lobster , snail , spider , worm |
|---|---|
| Task 2 | maple , oak , palm , pine , willow , orchids , poppies , roses , sunflowers , tulips , apples , mushrooms , oranges , pears , sweet peppers , cloud , forest , mountain , plain , sea |
| Task 3 | hamster , mouse , rabbit , shrew , squirrel , fox , porcupine , possum , raccoon , skunk , crocodile , dinosaur , lizard , snake , turtle , bear , leopard , lion , tiger , wolf |
| Task 4 | bicycle , bus , motorcycle , pickup truck , train , lawn mower , rocket , streetcar , tank , tractor , bed , chair , couch , table , wardrobe , bridge , castle , house , road , skyscraper |
| Task 5 | baby , boy , girl , man , woman , camel , cattle , chimpanzee , elephant , kangaroo , bottles , bowls , cans , cups , plates , clock , keyboard , lamp , telephone , television |

Table 27: Classes distribution of TinyImageNet separated by random seed 3456.

| Task 1 | Persian cat, cIIff, plunger, German shepherd, teddy, American lobster, hourglass, seashore, dumbbell, ice cream, nail, convertible, orangutan, coral reef, go-kart, king penguin, sulphur butterfly, lesser panda, kimono, comic book, cockroach, projectile, lakeside, chimpanzee, bannister, bucket, gondola, koala, IIfeboat, teapot, police van, pill bottle, hog, crane, cash machine, mushroom, water tower, black stork, ice lolly, scorpion |
|---|---|
| Task 2 | sewing machine, lemon, barn, Yorkshire terrier, stopwatch, lawn mower, thatch, pizza, barbershop, organ, computer keyboard, bighorn, cardigan, baboon, snail, syringe, spider web, Labrador retriever, pretzel, pomegranate, tarantula, pop bottle, trilobite, poncho, remote control, European fire salamander, altar, obelisk, binoculars, CD player, ladybug, miniskirt, cannon, wok, potter's wheel, cougar, chest, sunglasses, water jug, picket fence |
| Task 3 | rugby ball, steel arch bridge, refrigerator, espresso, dining table, monarch, brown bear, confectionery, beach wagon, scoreboard, flagpole, potpie, brass, bow tie, brain coral, backpack, chain, bison, pole, beer bottle, grasshopper, tailed frog, lion, torch, abacus, magnetic compass, standard poodle, goose, bullet train, African elephant, gazelle, triumphal arch, iPod, beacon, jinrikisha, fly, dugong, suspension bridge, ox, wooden spoon |
| Task 4 | Egyptian cat, volleyball, rocking chair, bullfrog, apron, swimming trunks, fountain, bikini, school bus, plate, guinea pig, oboe, maypole, goldfish, orange, drumstick, centipede, mashed potato, viaduct, military unIform, banana, sock, bathtub, guacamole, walking stick, pay-phone, alp, lampshade, bell pepper, meat loaf, tabby, tractor, sombrero, gasmask, frying pan, spiny lobster, jellyfish, sandal, vestment, snorkel |
| Task 5 | reel, basketball, parking meter, black widow, umbrella, trolleybus, Arabian camel, space heater, American alligator, albatross, sea cucumber, sea slug, cIIff dwelling, boa constrictor, mantis, freight car, Chihuahua, fur coat, beaker, moving van, barrel, acorn, caulIflower, birdhouse, academic gown, golden retriever, neck brace, candle, desk, bee, dam, punching bag, butcher shop, slug, dragonfly, limousine, sports car, turnstile, Christmas stocking, broom |

Table 28: Classes distribution of TinyImageNet separated by random seed 5678.

| Task 1 | reel, lemon, refrigerator, swimming trunks, stopwatch, lawn mower, German shepherd, flagpole, dumbbell, American alligator, backpack, cIIff dwelling, sulphur butterfly, kimono, trilobite, sock, bison, projectile, grasshopper, walking stick, lakeside, lion, pay-phone, bullet train, African elephant, birdhouse, gazelle, miniskirt, spiny lobster, pill bottle, hog, cougar, water tower, sunglasses, black stork, suspension bridge, ice lolly, broom, scorpion, picket fence |
|---|---|
| Task 2 | Egyptian cat, bullfrog, basketball, apron, Yorkshire terrier, monarch, pizza, guinea pig, umbrella, barbershop, drumstick, ice cream, nail, space heater, convertible, baboon, snail, orangutan, centipede, syringe, sea slug, mashed potato, military unIform, chain, cockroach, boa constrictor, tailed frog, academic gown, ladybug, golden retriever, cannon, neck brace, iPod, slug, crane, limousine, dugong, water jug, Christmas stocking, wooden spoon |
| Task 3 | sewing machine, cIIff, plunger, brown bear, teddy, oboe, maypole, orange, trolleybus, seashore, Arabian camel, brass, bighorn, brain coral, viaduct, king penguin, tarantula, lesser panda, comic book, poncho, mantis, remote control, chimpanzee, alp, lampshade, torch, tabby, IIfeboat, CD player, caulIflower, sombrero, frying pan, dam, beacon, jellyfish, cash machine, mushroom, jinrikisha, chest, ox |
| Task 4 | volleyball, rocking chair, steel arch bridge, espresso, thatch, black widow, school bus, plate, confectionery, beach wagon, American lobster, hourglass, computer keyboard, cardigan, coral reef, albatross, spider web, sea cucumber, beer bottle, bathtub, freight car, altar, beaker, bannister, magnetic compass, moving van, gondola, koala, binoculars, teapot, tractor, triumphal arch, gasmask, desk, bee, punching bag, potter's wheel, butcher shop, vestment, fly |
| Task 5 | rugby ball, Persian cat, barn, parking meter, dining table, fountain, bikini, organ, scoreboard, goldfish, potpie, bow tie, go-kart, Labrador retriever, pretzel, pomegranate, pop bottle, banana, pole, guacamole, Chihuahua, European fire salamander, obelisk, fur coat, bell pepper, bucket, abacus, meat loaf, barrel, standard poodle, goose, acorn, candle, police van, wok, sandal, dragonfly, snorkel, sports car, turnstile |

Table 29: Classes distribution of TinyImageNet separated by by grouping similar classes together.

| | |
|---|---|
| Task 1 | school bus,maypole,projectile,freight car,pay-phone,moving van,bullet train,birdhouse, tractor,triumphal arch,cannon,police van,crane,cash machine,jinrikisha,water tower, limousine,sports car,suspension bridge,turnstile,picket fence,refrigerator,barn,lawn mower, barbershop,beach wagon,scoreboard,flagpole,trolleybus,convertible,go-kart,viaduct, pole,bathtub,altar,obelisk,bannister,gondola,lIfeboat,bucket |
| Task 2 | reel,volleyball,rocking chair,basketball,plunger,parking meter,dining table,umbrella, oboe,hourglass,computer keyboard,space heater,backpack,pop bottle,beer bottle,remote control, lampshade,torch,abacus,barrel,CD player,teapot,candle,desk, frying pan,iPod,wok,potter's wheel,pill bottle,snorkel,sunglasses,water jug, broom,wooden spoon,rugby ball,sewing machine,stopwatch,plate,teddy,drumstick |
| Task 3 | Egyptian cat,bullfrog,German shepherd,brown bear,guinea pig,Arabian camel,baboon,Labrador retriever, king penguin,lesser panda,chimpanzee,tabby,goose,koala,gazelle,golden retriever, hog,cougar,black stork,ox,Persian cat,Yorkshire terrier,bighorn,orangutan, American alligator,bison,boa constrictor,Chihuahua,lion,standard poodle,African elephant,clIff, seashore,lakeside,alp,dam,steel arch bridge,fountain,clIff dwelling,magnetic compass |
| Task 4 | monarch,snail,albatross,spider web,sulphur butterfly,tarantula,cockroach,mantis, European fire salamander,ladybug,bee,slug,dragonfly,fly,scorpion,black widow, centipede,grasshopper,tailed frog,goldfish,coral reef,sea cucumber,spiny lobster,jellyfish, American lobster,brain coral,sea slug,trilobite,lemon,banana,guacamole,mushroom, thatch,orange,mashed potato,pomegranate,bell pepper,acorn,caulIflower,binoculars |
| Task 5 | cardigan,sock,fur coat,academic gown,miniskirt,neck brace,sombrero,gasmask, punching bag,sandal,Christmas stocking,apron,swimming trunks,bikini,bow tie,military unIform, kimono,poncho,espresso,pizza,organ,potpie,ice cream,nail, pretzel,beacon,butcher shop,vestment,chest,dugong,ice lolly,confectionery, brass,comic book,meat loaf,dumbbell,syringe,chain,walking stick,beaker |

