# OpenReview forum: "Towards Interpretable Continual Learning Through Controlling Concepts"
_ICLR.cc/2024/Conference — Submitted to ICLR 2024_

### Official Review · Reviewer_6BcB · 2023-11-04

**Soundness:** 3 good
**Presentation:** 3 good
**Contribution:** 3 good
**Rating:** 6
**Confidence:** 4

**Summary:**

This paper proposes a class incremental learning algorithm based on incrementally growing concept bottlenecks. It uses CLIP-dissect to identify neurons responsible for certain concepts and uses GPT to generate relevant concepts for a given class label. It freezes the part of the network that is responsible for previously learned concepts and adds new concepts for new classes. Lastly, the network maps from concepts to classes like in Concept Bottleneck Models.

**Strengths:**

- Using interpretable concepts as middle points to guide through incremental class learning is an interesting idea.
- Using pretrained models (backbone, CLIP, GPT) to assist continual learning is a novelty.
- Experimental results show that the proposed method is superior to other continual learning algorithms.

**Weaknesses:**

- Since the paper utilize a pretrained backbone, there is not much difference between the proposed method and the baselines. Moreover, it is unclear whether the gain comes from its continual learning ability or just the concept bottleneck. It would be good to see whether the proposed GPT+Concept Bottleneck procedure works well for a non-incremental learning setting.
- The paper is most related to DEN, but there is no comparison to the method. The paper could be compared to DEN by having the same pretrained backbone network with additional two layers learned by DEN instead of incremental concept bottlenecks.
- The paper lacks clarity on the GPT concept generation and filtering procedure. It would be helpful to give examples on what the concepts are (move some figures from Appendix to main text). It is also important to share the text prompts used to generate concepts.

**Questions:**

N/A

---

> ### Author Response · Authors · 2023-11-19
> **Author Response**
>
> Thank you for your positive feedback! We are glad to learn that you think our work is interesting and novel. Below we provide a few more details in response to your comments and requests.
>
> **#1 Details of CC-CBM**
>
> We describe the CC-CBM in Section 4 (p.5 -p.7). Even though it uses a pretrained backbone, we think it is very different from other baselines [2,3,4]. We combine CC with LF-CBM [1] to build CC-CBM. CC is for its continual learning ability. To know where performance gain comes from, we need to investigate CC-CBM in two steps:
> 1. *Finetune standard model v.s. Finetune CBM* : To understand the impact of the concept bottleneck.
> 2. *Finetune CBM v.s. CC-CBM* : To understand the impact of CC.
>
> Both points have been done in our paper, please see the main results (Section 5.1 from p.7- p.9) and Appendix (p.23 - p.26):
> * Overall, Finetune CBM outperforms Finetune standard model by up to 2.3% in $\bar{A}_T$ and up to 16.4% in $\bar{F}_T$.
> * CC-CBM makes CBM’s continual learning ability even better, as it outperforms Finetune CBM by up to 1.8% in $\bar{A}_T$ and up to 12.4% in $\bar{F}_T$.
>
> The non-incremental learning setting is studied in LF-CBM, which shows comparable performance to the standard model. In contrast, in this paper we have studied the GPT + concept bottleneck in the continual learning setting, which shows that this direction is promising and is a novel solution to address the catastrophic forgetting as shown in Section 5.1 of the main paper (p.7- p.9).
>
> **#2 Comparison with DEN**
>
> Following your instructions, we compare CC-CBM (ours) and Finetune-CBM (ours) with DEN in below Table R1. Our methods outperform DEN by 15.07% and 15.81% in $\bar{A}_T$ respectively.
>
> Table R1: Experiment results on CIFAR-100 in a 5-task scenario.
>
> |              | $\bar{A}_T$ |
> | ------------ | ------------ |
> | DEN          | 9.18         |
> | Finetune-CBM (Ours) | **24.99**        |
> | CC-CBM (Ours)      | 24.25        |
>
>
>
> **#3 GPT-3 usage and example**
>
> CC-CBM is builded upon LF-CBM [1]. It follows the procedure described in LF-CBM to use GPT-3. Example prompts:
> ```
> • List the most important features for recognizing something as a {class}:
> • List the things most commonly seen around a {class}
> • Give superclasses for the word {class}:
>
> ```
>
>
> **#4 Summary**
>
> To summarize, we have:
> 1. Discussed in **#1** on the details of CC-CBM.
> 2. Provided in **#2** for the comparison with DEN.
> 3. Discusses in **#3** on the GPT-3 usage.
>
> We believe we have addressed all your concerns. Please let us know if you still have any reservations and we would be happy to address them.
>
> Reference:
>
> [1] Label-free Concept Bottleneck Models. ICLR 2023
>
> [2] Self-Sustaining Representation Expansion for Non-Exemplar Class-Incremental Learning. CVPR 2022
>
> [3] Training networks in null space of feature covariance for continual learning. CVPR 2021
>
> [4] Gradient episodic memory for continual learning. NeurIPS 2017

---

> > ### Author Response · Authors · 2023-11-22
> > **Request Rebuttal feedback from Reviewer 6BcB**
> >
> > Dear Reviewer 6BcB,
> >
> > Following your request, we have provided more details with additional experiment comparison with DEN.
> >
> > We believe we have addressed all your concerns. Please let us know if you still have any reservations and we would be happy to discuss further!

---

> ### Comment · Reviewer_6BcB · 2023-12-04
>
> I thank the reviewer for their responses. The added results of DEN are good, and it would be good to add DEN to other datasets too.
> While I think the idea is interesting and results are significant, I also share some of the concerns by other reviewers that the method could suffer from its complexity and the writing could also be clearer to facilitate understanding. I remain my current score rating of the paper.

---

### Official Review · Reviewer_zdWA · 2023-11-06

**Soundness:** 2 fair
**Presentation:** 2 fair
**Contribution:** 2 fair
**Rating:** 5
**Confidence:** 4

**Summary:**

The paper proposes a framework using CLIP Dissect to alleviate catastrophic forgetting in continual learning by controlling concepts. A neuron is denoted as a “concept unit” if it activates highly, and is hence highly correlated with, a human-understandable concept. These concepts are architecturally added, frozen and reused in subsequent tasks. A continual extension to concept bottleneck models is also presented, which builds on top of Label Free-CBMs.

**Strengths:**

Neuron-level interpretability used for continual learning is an underexplored direction. The paper builds upon several existing works like CLIP Dissect and LF CBMs, using them in a continual setting. The notion of concept evolution was quite interesting. Additionally, the background and related work sections are covered well. I also appreciate the detailed experimental studies presented in the Appendix.

**Weaknesses:**

**Scalability**
There is no analysis provided on the order of the number concepts added to Ct for new tasks. This would affect how scalable the method is, especially as it is mentioned that repeated concepts in subsequent tasks are added to the concept set as well.

**Motivation**
The method does not seem to necessitate having interpretable concepts to alleviate catastrophic forgetting. The same thing could have been carried out on the classification layer itself using a vision-language aligned model and backbone. To be more specific, the entire dissection and subnetwork search process could have been performed directly on the classes. How are concepts or rather interpretability in general helping here? The paper seems to be attempting to address two different albeit related things, although the motivation for doing so is not very clear.

**Formulation**
* By design, it appears that the proposed method can only be used for CNN models and not transformer architectures. It would be nice to see how the proposed work can be more contemporary in its application.
* In the freeze-all variant, it is possible that classes in newer tasks may be based on concepts that were available earlier. How would the model learn these associations if the weights for old tasks are not allowed to change?

**Experiments**
* The paper shows experiments on relatively small datasets. Related to my point on scalability, I would like to see some results on larger scale datasets.
* The baselines for CL are not contemporary – there have been several state-of-the-art baselines for CL in the last 2 years, which are not considered. Additionally, no existing continual interpretable baselines have been included like ICICLE (ICCV 2023). (While I understand that the ICCV conference happened after the ICLR deadline, this work was available on arXiv since March 2023, https://arxiv.org/abs/2303.07811)
* I would also like to see some analysis on other vision-language aligned models like FLAVA.
* It would have been nice to see some discussion on subnetworks beyond Sec 3. How big were the learned subnetworks? How many weights were actually frozen on the different datasets?

**Presentation**
The writing is unclear in a few places. For example:
* “it’s not considering classification accuracy of CBM in continual learning setting, which is different than our goal.” (pg 5) and “the Concept Controller strategy follows the similar idea as CC’s in step 4” (pg 6). It is difficult to understand what is trying to be conveyed in these sentences.
* In Fig 3, it is not clear whether the network is from top to bottom or the other way, since there are no arrows. This makes it hard to understand the two schemes.
* In Sec 3, the paper states that the subnetwork is frozen. In Sec 4, it states that the concepts are frozen. Is a concept a neuron or a sub-network? This is unclear.
* Since the main premise of this work is on concepts and their subsequent use of interpretability, it would have been nice to see results such as Figures 6 and 7 in the main paper. The primary results in the main paper are all standard CL metrics. Note that Tables 3 (and 8 in the Appendix) only studies the concept consistency – it does not study interpretability.

**Questions:**

1. On expanding the concept set in successive tasks, it is stated that existing concepts are also added to the current concept set as they could capture a different context. Please clarify how this context is captured.
2. How does the freezing strategy of concept controller account for old concepts occurring in new classes?
3. How does the framework scale to large datasets?
4. Other than the fact that a neuron-level interpretable model is being used, is such a model even necessary to the problem the paper attempts to address? As the same purpose could be served by using any VL-aligned model.

---

> ### Author Response · Authors · 2023-11-19
> **Author Response (1/2)**
>
> Thank you for your valuable feedback, please see our reply below.
>
> **#1 Scalability**
> 1. Baseline papers [1, 2, 3] mostly focus on the CIFAR dataset. TinyImagenet is the largest dataset in all baseline papers we compared except SSRE [1] which does experiments in ImageNet-100. Due to the rebuttal constraint, we have tried to add additional experiments on large scale dataset (e.g. ImageNet-10), and the results are in Table R1 and R2. The experiment results are still under the same trend: CC outperforms other exemplar-free methods by up to 0.87% in $\bar{A}_T$ and up to 5.77% in $\bar{F}_T$. Meanwhile, can improve exemplar-based methods by combining them by up to 1.83% in $\bar{A}_T$ and up to 2.87% in $\bar{F}_T$.
> 2. In Table R3 and Table R4, we provide the number of concepts added in the concept bottleneck layer (CBL) for CC-CBM.
> 3. For large-scale datasets in common continual learning benchmarks, our methods perform well, as shown Table 1 and Table 2 (p.8 -p.9) in the draft and the new ImageNet-10 experiments. Due to the expansion ability of CC-CBM, we believe that we can handle other large-scale datasets as well.
>
> Table R1: Experiment results on ImageNet-10 in a 5-task scenario.
> |                    | $\bar{A}_T$ | $\bar{F}_T$ |
> | ------------------ | ------------ | ------------ |
> | Finetune           | 23.33        | 81.20        |
> | EWC                | 24.14        | 78.46        |
> | CC_sol0 (Ours)     | **25.01**        | **77.20**        |
> | GEM                | 31.34        | 61.71        |
> | CC_sol0_GEM (Ours) | **33.17**        | **59.65**        |
>
> Table R2: Experiment results on ImageNet-10 in a 5-task scenario, the backbones are pretrained on Place365 dataset.
> |                   | $\bar{A}_T$ | $\bar{F}_T$ |
> | ----------------- | ------------ | ------------ |
> | Finetune          | 23.83        | 81.92        |
> | EWC               | 25.40        | 78.98        |
> | CC_CBM (Ours)     | **26.12**        | **73.21**        |
> | GEM               | 32.66        | 60.27        |
> | CC_CBM_GEM (Ours) | **34.37**        | **57.40**        |
>
> Table R3: Average concepts added per task in a 5-task scenario.
> |                   | CIFAR-10 | CIFAR-100 | CUB200 | ImageNet-10 | TinyImageNet |
> | ----------------- | -------- | --------- | ------ | ----------- | ------------ |
> | Number of Concept | 37.4     | 133.1     | 276.6  | 35.6        | 301.8        |
>
> Table R4: Average concepts added per task in a 10-task scenario.
> |                   | CIFAR-100 | TinyImageNet |
> | ----------------- | --------- | ------------ |
> | Number of Concept | 154.6     | 284.3        |
>
> **#2 Motivation for the role of concepts and interpretability**
>
> We agree that the subnetwork search process can be performed on the classification layer directly, just like DEN [2]. However, this gives us no idea about what knowledge and information is learned and preserved in the model, which lacks interpretability. This is the main motivation of our proposed method. In the detailed motivation (please see **#5** in the **Author Response (2/2)** to **Reviewer Hm1u**), we saw the catastrophic forgetting of concepts in other baselines, which motivated us to design the proposed Concept Controller.  As experiment results in the main results (Section 5.1 from p.7- p.9) and Appendix (p.23 - p.26) show, controlling concepts improves the model’s performance in continual learning. Moreover, our methods provide interpretability, which is different from other continual learning algorithms. Finally, We added a comparison between our method and DEN, please see **#2** in the **Author Response** to **Reviewer 6BcB**.
>
> **#3 Questions regarded to vision-language aligned (VL-aligned) model**
> 1. We build CC-CBM based on LF-CBM [4]. Following LF-CBM’s procedure, we use CLIP to calculate the activation matrix $P$ as described in Section 2.2 (p.3). We can replace CLIP with other VL-aligned models as long as they have a text encoder and an image encoder to calculate matrix $P$. We would like to point out that the choice of the VL-aligned model is related to LF-CBM’s design instead of the continual learning problem. We will add this discussion in the revision and leave it as a future work.
> 2. As we discussed in **#2**, controlling concepts provides interpretability and improves performance. Just using a VL-aligned model is not sufficient to understand the concepts inside itself.

---

> ### Author Response · Authors · 2023-11-19
> **Author Response (2/2)**
>
> **#4 Recent baselines in experiments**
> 1. We have compared the two baselines within two years: Adam-NSCL[3] and SSRE [1]. As you requested, we compare our methods with ICICLE, and the experiment results are in Table R5. Our method outperforms ICICLE by 2.02% in $\bar{A}_T$ and 31.68% in $\bar{F}_T$.
> 2. ICICLE focuses on part-based prototype concepts. Our work focuses on text-based concepts instead, which allows more general interpretability. Meanwhile, it is only suitable for particular model architectures whereas our methods are suitable for any CNN-based models.
>
> Table R5: Experiment results on CUB200 [5] in a 5-task scenario. The backbones are pretrained on Place365 dataset.
> |               | $\bar{A}_T$ | $\bar{F}_T$ |
> | ------------- | ------------ | ------------ |
> | Finetune      | 10.16        | 60.73        |
> | EWC           | 11.02        | 62.43        |
> | LwF           | 12.50        | 64.50        |
> | Adam-NSCL     | 13.27        | 40.73        |
> | ICICLE        | 11.40        | 53.66        |
> | CC-CBM (Ours) | **13.42**        | **21.98**        |
>
> **#5 Details for freeze concepts and subnetwork**
> 1. In CC, classes in new tasks may reuse old concepts learned in previous classes. CC freezes the subnetwork from the concept units to the bottom, but does not freeze the associations between the classification layer and concept units. Therefore, the new classes can reuse old concepts. We added new experiments on CUB200 [5], which is a bird species classification dataset and has many concepts overlapped between classes. We measured the forward transfer metric [9] (FWT) to understand whether old concepts help new classes. The experiment results are in Table R6, which shows our method can reuse concepts efficiently and improve a model’s performance.
> 2. We added the subnetwork size of different datasets in appendix A.14 (p.29). Overall, a more complicated dataset has a bigger subnetwork.
>
> Table R6: FWT on comparing our methods with baselines.
> | CUB200, 5 tasks            | FWT     |
> | -------------------------- | ------- |
> | Without memory buffer:     |
> | Finetune                   | 0.76    |
> | EWC                        | 3.83    |
> | SI                         | 13.50   |
> | LwF                        | 10.20   |
> | CC-freeze-all (Ours)       | 28.79   |
> | CC-freeze-part  (Ours)     | **30.07**   |
> | With memory buffer:        |
> | GEM                        | 1.35    |
> | CC-freeze-all-GEM  (Ours)  | **26.76**   |
> | CC-freeze-part-GEM  (Ours) | 26.68   |
> | MIR                        | \-12.44 |
> | CC-freeze-all-MIR  (Ours)  | 6.89    |
> | CC-freeze-part-MIR  (Ours) | 8.22    |
>
> **#6 Generalization to transformers**
>
> We would like to point out that whether concepts can be disentangled in transformers is still under research [6, 7, 8]. We believe this is an interesting future work after transformers’ interpretability has made solid progress.
>
> **#7 Presentation**
> 1. Thank you for pointing out the unclear part of our paper. We have revised them in the revision.
> 2. In Section 3, the subnetwork of a concept unit is frozen. In Section 4, we freeze the concept unit itself. Concept unit is a neuron as we stated in Section 1 (p.1).
> 3. Due to the page limit, we are not able to put Figures 6 or 7 in the main paper. The discussion for CC’s interpretability is in Section 3 (p.5).
>
> **#8 Summary**
> To summarize, we have:
> 1. Discussed in **#1** on our methods’ scalability, and add a new experiment on a large-scale dataset.
> 2. Discussed in **#2** on the role of concepts and interpretability.
> 3. Discussed in **#3** on questions related to vision-language aligned models.
> 4. Discussed in **#4** on the latest baselines we have compared, and add a comparison with ICICLE.
> 5. Described in **#5** on the mechanism and details of subnetwork.
> 6. Described in **#6** on the generalization to transformers.
> 7. Discussed in **#7** on the presentation of our paper.
>
> We believe we have addressed all your concerns. Please let us know if you still have any reservations and we would be happy to address them.
>
> Reference:
>
> [1] Self-Sustaining Representation Expansion for Non-Exemplar Class-Incremental Learning. CVPR 2022
>
> [2] Lifelong Learning with Dynamically Expandable Networks. ICLR 2018
>
> [3] Training networks in null space of feature covariance for continual learning. CVPR 2021
>
> [4] Label-free Concept Bottleneck Models. ICLR 2023
>
> [5] The caltech-ucsd birds-200-2011 dataset. 2011.
>
> [6] An interpretability illusion for bert. arXiv:2104.07143 2021
>
> [7] "Toy Models of Superposition", Transformer Circuits Thread, 2022.
>
> [8] "Towards Monosemanticity: Decomposing Language Models With Dictionary Learning", Transformer Circuits Thread, 2023.
>
> [9] Gradient episodic memory for continual learning. NeurIPS 2017

---

> > ### Author Response · Authors · 2023-11-21
> > **Request Rebuttal feedback from Reviewer zdWA**
> >
> > Dear Reviewer zdWA,
> >
> > Following your requests, we have provided additional experiment results on large-scale dataset, comparison with the latest baselines ICICLE and forward transfer metrics. We have also clarified your multiple questions in the response. Please see General response and Appendix A. 13, A.14 on p.28-29 in revised draft
> >
> > We believe that we have addressed all your concerns and with additional new experiments and clarifications. Please let us know if you still have any reservations and we would be happy to address them. Thank you!

---

> > > ### Author Response · Authors · 2023-11-22
> > > **Gentle Reminder on requesting rebuttal feedback from Reviewer zdWA**
> > >
> > > Dear Reviewer zdWA,
> > >
> > > As the rebuttal period is ending in one day, we would like to follow up with you on the feedback to our rebuttal response.
> > >
> > > We believe that we have addressed all your concerns and with additional new experiments and clarifications. Please let us know if you still have any reservations and we would be happy to address them.
> > >
> > > Thank you!

---

> ### Author Response · Authors · 2023-11-23
> **Polite reminder on requesting for final rebuttal feedback from Reviewer zdWA**
>
> Dear Reviewer zdWA,
>
> As the rebuttal period is ending within 7 hrs, we would like to ask for the feedback to our rebuttal response.
>
> As we did not hear from you, we are not sure if you have additional questions or comments. We believe that we have addressed all your concerns and with additional new experiments and discussions (see **General response** and Appendix A. 13, A.14). Please let us know if you still have any reservations and we would be happy to address them.
>
> Thank you!

---

### Official Review · Reviewer_Hm1u · 2023-11-06

**Soundness:** 3 good
**Presentation:** 3 good
**Contribution:** 2 fair
**Rating:** 3
**Confidence:** 3

**Summary:**

The paper proposes a continual learning pipeline with quite a few modules (such as dissection-based continual training and concept bottleneck). The training also contains multiple steps. The final empirical results showcase its benefits compared to buffer-free continual learning methods.

**Strengths:**

- The paper is clearlly written and generally easy to follow.

- The idea of introducing concept bottlenecks to continual learning is interesting and worth exploring.

- The experimental results look good on the benchmark datasets.

**Weaknesses:**

- I find the proposed method quite complex and ad-hoc in general. The dissection-based continual training is interesting, but it is eseentially to incorporate the dissection into [1].

- The introduction of label-free concept bottleneck to the proposed framework makes it even more complex and also difficult to find which part actually contributes to the performance gain. Therefore, an ablation study has to be performed. What if we combine label-free concept bottleneck to DER directly. How does it perform? The paper needs to study each added module carefully and show its advantages.

- The motivation to design such a complex system is weak. The usage of label-free concept bottleneck will introduce additional information from GPT-3, which is questionable. One can easily achieve good performance if you use store the text label and perform zero-shot classification on continual learning dataset (which can easilly outperform your results). Even if you use label-free concept bottlenecks, the addtional text information is still leaked to your model. I am not sure whether this is still a fair comparison.



[1] Der: Dynamically expandable representation for class incremental learning, CVPR 2021

**Questions:**

See the weakness section

---

> ### Author Response · Authors · 2023-11-16
> **Author Response (1/2)**
>
> Thank you for your valuable feedback, please see our reply below.
>
> **#1 Difference between our methods and DER (CVPR 2021)**
>
> We would like to highlight our contributions when comparing with DER (CVPR 2021):
> 1. We agree that architecture-based methods like DER are explored in other works, as we discussed in Section 2.1 (p.3). However, we are the first one to introduce neuron-level interpretability to continual learning, which is non-trivial. We are happy to hear your comment that our work in this direction is interesting and worth-exploring.
> 2. As we mentioned in the Introduction session (p.2), interpretability allows us to improve the "method design" in an interpretable manner. This is unexplored in DER and other architectured-based methods.
> 3. DER requires a different feature extractor for each task to retain learned knowledge. On the other hand, our methods use the same feature extractor to learn all tasks, which is more efficient.
>
> **#2 Ablation Study for CC-CBM**
>
> We have done ablation studies for CC-CBM in the main results and appendix:
> 1. When introducing label-free CBM (abbreviated as LF-CBM), we compare the Finetune strategy vs CC strategy in Step 2 (p.6), the experiment results are in the main results (Section 5.1 from p.7- p.9) and Appendix (p.23 - p.26).
> 2. Ablation studies for freezing $W_C$ and regularizing $W_F$ are in Appendix A.5 (p.17).
>
> These ablation studies have shown the contribution of each part in CC-CBM. Meanwhile, we believe that combining LF-CBM with DER (CVPR 2021) may not be a fair ablation study for our CC-CBM for the following reasons:
> 1. DER requires a new backbone for each task, while CC-CBM only needs one.
> 2. DER official code is not complete, since the code to learn a new backbone is missing. The issues have been complained about in their official codebase (https://github.com/Rhyssiyan/DER-ClassIL.pytorch/issues/18).
> 3. DER is an exemplar-based method, while CC-CBM is an exemplar-free method. We have compared with other exemplar-free methods (EWC, SI, LwF, Adam-NSCL, SSRE) and exemplar-based methods (GEM, MIR, DER-NeurIPS-2020 [11]). Please see Section 5.1 in p.7 for references. Note that the DER-NeurIPS-2020 is a different method published in [11] but happens to have the same name “DER” as the method you suggested (CVPR 2021) and hence we denote [11] as DER-NeurIPS-2020. Overall, our methods outperform other exemplar-free baselines and improve exemplar-based methods when combining them.
> 4. We have compared with baseline methods that are more recent than DER, e.g. SSRE (CVPR 2022) [10], and show that our results outperforms it in $L_T$ by up to 57.1% in Table 18 (p. 27, appendix A.12). Unfortunately, SSRE did not compare with DER either.
>
> We will add the above discussion into the revision. Thank you for your suggestions!
>
> **#3 Additional Information from GPT-3**
>
> We would like to highlight several points for the potential issue:
> 1. Using GPT-3 to general concept sets is firstly proposed from an earlier work LF-CBM [7], which has many concurrent and follow-up works that use language models to create concept bottleneck layers, e.g. see [8, 9]. In fact, introducing additional information is widely exploited when designing CBM [1, 2, 3], and is reasonable task-relevant information that we can leverage.
> 2. We agree that the potential leaking of CBM is an interesting problem, but we would like to point out that this is not the focus of our work as this is an universal problem in CBM. We will leave it as a future work and add discussion in the revised draft.
> 3. Using additional information to enhance a model's performance is common in other fields as well [4, 5].
> 4. Our method is still under the setting of continual learning because during the training phase, the information for each class only appears in one task.
>
> Thus, we believe that our methods are fair in the continual learning setting.

---

> ### Author Response · Authors · 2023-11-16
> **Author Response (2/2)**
>
> **#4 Further information for additional experiments**
>
> Following your suggestion, we performed the zero-shot classification experiment. We found that zero-shot classification on continual learning will lead to bad performance. In the 5-task scenario, we pretrained the backbone only on the first task, and did zero-shot classification for the following tasks. The experiment results of zero-shot classification are much worse than our proposed CC-CBM as below Table R1 shows.
>
> Table R1: Comparison between zero-shot classification and our methods, in the 5-task scenario of CIFAR-100. The experiment results shows that our methods (CC-CBM) outperforms the zero-shot verison in average accuracy $A_T$ and learning accuracy $L_T$ (ability to learn new tasks).
> |                  | $A_T$         | $L_T$         |
> | ---------------- | ------------- | ------------- |
> | Zero-shot CC-CBM | 4.34          | 15.36         |
> | CC-CBM           | **16.37** | **62.33** |
>
> **#5 Motivation**
>
> We have briefly discussed our motivation in the Introduction section (p.1). Overall, our goal is to design continual learning algorithms which are interpretable. Previous work [6] points out the benefits of human-interpretable concepts for transfer learning.
>
> Meanwhile, we measured the evolution of human-understandable concepts in the models when learning sequences of tasks. The experiment results in Table 3 in main draft (p.9) and Table 9 and Table 10 in Appendix p.19) show that existing continual learning algorithms don’t really preserve human-understand concepts. To prevent this behavior, we think it is reasonable to design an algorithm that aims to mitigate catastrophic forgetting of concepts, which is the main motivation for us to design the proposed Concept Controller. We will add this discussion to the introduction in revision to make the motivation more clear.
>
> **#6 Summary**
>
> To summarize, we have:
> * Clarified in **#1** on our contributions when comparing with DER (CVPR 2021).
> * Described  in **#2** on the ablation studies we have done, and why comparing with DER is not a fair ablation study.
> * Discussed in **#3** about the additional information from GPT-3.
> * Provided in **#4** for the additional experiments as requested.
> * Clarified in **#5** on our motivation.
>
> We believe we have addressed all your concerns. Please let us know if you still have any reservations and we would be happy to address them.
>
>
> Reference:
>
> [1] Post-hoc Concept Bottleneck Models. ICLR 2023
>
> [2] Interpretable Basis Decomposition for Visual Explanation. ECCV 2018
>
> [3] Interpretability beyond classification output: Semantic bottleneck networks. arXiv:1907.10882, 2019
>
> [4] Compare to The Knowledge: Graph Neural Fake News Detection with External Knowledge. ACL 2021
>
> [5] Entity-Aware Dual Co-Attention Network for Fake News Detection. EACL 2023 Findings
>
> [6] Network dissection: Quantifying interpretability of deep visual representations. CVPR 2017.
>
> [7] Label-free Concept Bottleneck Models. ICLR 2023
>
> [8] Language in a Bottle: Language Model Guided Concept Bottlenecks for Interpretable Image Classification. CVPR 2023
>
> [9] Learning Concise and Descriptive Attributes for Visual Recognition. ICCV 2023
>
> [10] Self-Sustaining Representation Expansion for Non-Exemplar Class-Incremental Learning. CVPR 2022
>
> [11]  Dark experience for general continual learning: a strong, simple baseline. NeurIPS 2020

---

> ### Author Response · Authors · 2023-11-20
> **Request Rebuttal feedback from Reviewer Hm1u**
>
> Dear Reviewer Hm1u,
>
> We believe that we have addressed all your concerns and with additional new experiments (see General response and Appendix A. 13, A.14). Please let us know if you still have any reservations and we would be happy to address them. Thank you!

---

> > ### Author Response · Authors · 2023-11-22
> > **Gentle reminder on requesting rebuttal feedback from Reviewer Hm1u**
> >
> > Dear Reviewer Hm1u,
> >
> > As the rebuttal period is ending in one day, we would like to follow up with you on the feedback to our rebuttal response.
> >
> > We believe that we have addressed all your concerns and with additional new experiments (see General response and Appendix A. 13, A.14). Please let us know if you still have any reservations and we would be happy to address them.
> >
> > Thank you!

---

> ### Author Response · Authors · 2023-11-23
> **Polite reminder on requesting for final rebuttal feedback from Reviewer Hm1u**
>
> Dear Reviewer Hm1u,
>
> As the rebuttal period is ending within 7 hrs, we would like to ask for the feedback to our rebuttal response.
>
> As we did not hear from you, we are not sure if you have additional questions or comments. We believe that we have addressed all your concerns and with additional new experiments and discussions (see **General response** and Appendix A. 13, A.14). Please let us know if you still have any reservations and we would be happy to address them.
>
> Thank you!

---

### Author Response · Authors · 2023-11-19
**General Response: Overview of new results**

In response to all reviewer comments, we have conducted several new experiments and made some changes to existing manuscript content. New text is written in red for visibility in the PDF.

**# New experiments**

1. We have compiled all the new results and discussion in appendix A.13 and A.14 (p.28-p.29) in the PDF with a below short summary:
In Appendix A.13, we compare CC-CBM with ICICLE [1], a recent interpretable continual learning work. The experiment result in Table 20 (p.28) shows that our method outperforms ICICLE by 2.02% in $\bar{A}_T$ and 31.68% in $\bar{F_T}$. Meanwhile, we add a description of ICICLE in Section 2.1 (p.3).
2. In Appendix A.14, we discuss the details of the Concept Controller. For CC, we show the subnetwork sizes in different datasets in Figure 8 (p.29). Generally, a more complicated dataset has a bigger subnetwork, but the size is adjustable by tuning hyperparameters. For CC-CBM, we show the number of concepts added per task in Table 21 and Table 22 (p.29). Meanwhile, we discuss two properties related to LF-CBM: (1) analysis of the vision language aligned model and (2) the potential leakage problem of LF-CBM.



**# Other Changes**
1. Provide detailed motivation in Section 1 (p.2).
2. Clarified the description of other CBM works in Section 2.2 (p.4).
3. Make Figure 3 (p.5) clearer by pointing out the direction.
4. Clarified the procedure of CC-CBM in Section 4 (p.6).

Reference:

[1] Icicle: Interpretable class incremental continual learning. ICCV 2023

---

### Meta-Review · Area_Chair_xDUx · 2023-12-08

**Metareview:**

(a) Summarize the scientific claims and findings of the paper based on your own reading and characterizations from the reviewers.
- The authors propose a new idea for preventing catastrophic forgetting in continual learning
- The idea is to identify and preserve "concept neurons," a term borrowed from the transfer learning literature
- The authors instantiate two methods, one using regularization and the other a growing architecture. The methods are also combined.
- Both methods seem to provide empirical gains (improved accuracy and reduced forgetting)

(b) What are the strengths of the paper?
- There are elements of novelty within the proposed methods (including the use of concept bottlenecks and neuron-level interpretability
- The empirical evaluation on standard datasets is sound
- The paper is overal clearly written and easy to follow

(c) What are the weaknesses of the paper? What might be missing in the submission?
- The method was described as being complex by several reviewers and as a combination of prior work (LF-CBM and DER)
- Evaluation uses standard but small-scale datasets (but new larger-scale studies were done during the rebuttal period)
- The exact interaction
- The empirical gains show nuances that are not discussed (e.g., in Table 1, some of the improvements might not be statistically significant)

**Justification For Why Not Higher Score:**

The initial reviews assessed the paper as somewhat borderline, with two of the three reviewers initially suggesting it was below the acceptance threshold.

The authors responded to reviewers and were very active in soliciting replies from the reviewers. Two of the three reviewers ended up providing a response (one sent privately to the AC and other reviewers). Both recognized that the author's response was useful, but their overall assessment of the work, its main strengths and weaknesses remain.

Reviewer zdWA suggests that the work, while having elements of novelty, essentially combines two previous baselines (DER and LF-CBM). The reviewer as as well as reviewer 6BcB also mentioned the perceived complexity of the proposed approach. The empirical study does not support the value of the contribution. The study is well designed, but is done on standard datasets and reports some improvements.

The value of using GPT-3 for such a task was also discussed. One one hand, the authors report weaker zero-shot results in their discussion. On the other, it is intriguing to use a large-scale pre-trained language model to help classify CIFAR images.

Reviewer Hm1u, mentioned the need to study larger-scale datasets. The authors report new results in the response, but the no variance estimate is provided making it difficult to conclude about the magnitude of the potential gains.

As written above, providing more nuance around the discussion of the results also seems warranted given the reported variations.

Overall, the combination of these weaknesses along with the overall assessment of the three reviewers indicate that this paper is not yet ready to be published at this venue. I understand that this might be frustrating to the authors and encourage them to pursue this interesting line of work. I hope some of these comments will be helpful.

**Justification For Why Not Lower Score:**

N/A

---

### Decision · Program_Chairs · 2024-01-16

Reject